# Bayesian Integration of Astrochronology and Radioisotope Geochronology

[1,2,*]Robin B. Trayler, [3]Stephen R. Meyers, [4]Bradley B. Sageman, [2]Mark D. Schmitz

[1]Department of Life and Environmental Sciences, University of California, Merced, CA
[2]Department of Geosciences, Boise State University, Boise ID
[3]Department of Geosciences, University of Wisconsin, Madison, WI
[4]Department of Earth and Planetary Sciences, Northwestern University, Evanston, IL

[*]Corresponding author: rtrayler@ucmerced.edu

## Abstract

Relating stratigraphic position to numerical time using age-depth models plays an important role in determining the rate and timing of geologic and environmental change throughout Earth history. Astrochronology uses the geologic record of astronomically derived oscillations in the rock record to measure the passage of time and has proven a valuable technique for developing age-depth models with high stratigraphic-temporal resolution. However, in the absence of anchoring dates, many astrochronologies float in numerical time. Anchoring these chronologies relies on radioisotope geochronology (e.g., U-Pb,
$^{40}Ar/^{39}Ar$), which produces high-precision (<±1%), stratigraphically distributed point estimates of age.

In this study, we present a new R package, astroBayes for a Bayesian inversion of astrochronology and radioisotopic geochronology to derive age-depth models. Integrating both data types allows reduction in uncertainties related to interpolation between dated horizons, and the resolution of subtle changes in sedimentation rate, especially when compared to existing Bayesian models that use a stochastic random walk to approximate sedimentation variability. The astroBayes inversion also
incorporates prior information about sedimentation rate, superposition, and the presence/ absence of major hiatuses. The resulting age-depth models preserve both the spatial resolution of floating astrochronologies, and the accuracy and precision of modern radioisotopic geochronology.

We test the astroBayes method using two synthetic data sets designed to mimic real-world stratigraphic sections. Model uncertainties are predominantly controlled by the precision of the radioisotopic dates, and are relatively constant with depth
while being significantly reduced relative to "dates-only" random walk models. Since the resulting age-depth models leverage both astrochronology and radioisotopic geochronology in a single statistical framework they can resolve ambiguities between the two chronometers. Finally, we present a case study of the Bridge Creek Limestone Member of the Greenhorn Formation where we refine the age of the Cenomanian-Turonian Boundary, showing the strength of this approach when applied to deep-time chronostratigraphic questions.

## 1    Introduction

Linking the rock record to numerical time is a crucial step when investigating the timing, rate, and duration of geologic, climatic, and biotic processes, but constructing chronologies (age-depth modeling) from the rock record is complicated by a variety of factors. The premier radioisotopic geochronometers enable direct determination of a numerical date from single mineral crystals (e.g., sanidine, zircon) to better that 0.1% throughout Earth history (Schmitz and Kuiper, 2013). However, rocks amenable to radioisotopic dating, mostly volcanic tuffs, may only occur as a few dispersed horizons within a stratigraphic section. This leads to the problem of a small number of high-precision dates scattered throughout stratigraphy with limited chronologic information between these horizons. Consequently, chronologies developed using only radioisotopic dates have widely varying uncertainties throughout a given stratigraphic record, with precise ages near the position of the dates and increasing uncertainties with distance from the dated horizons (Blaauw and Christen, 2011; Parnell et al., 2011; Trachsel and Telford, 2017; Trayler et al., 2020).

Adding more chronological information is the best way to improve age-depth model construction (Blaauw et al., 2018). In particular, including stratigraphically continuous data can significantly reduce model uncertainties. Astrochronology uses the geologic record of oscillations in Earth's climate system ("Milankovitch cycles") to measure the passage of time in strata (Hinnov, 2013; Laskar, 2020). Some of these oscillations can be linked to astronomical physics with well understood periods, including changes in the ellipticity of Earth's orbit (eccentricity; ~0.1 Ma, 0.405 Ma), Earth's axial tilt (obliquity; ~0.041 Ma), and axial precession (precession; ~0.02 Ma) (Laskar, 2020). The manifestation of these astronomical periods in the rock record can be leveraged as a metronome that provides a direct link between the rock record and time (either "floating" or "anchored" astrochronologies"; see reviews of Hinnov (2013) and Meyers (2019)). Unlike radioisotopic dating methods, astrochronology produces near-continuous chronologies from stratigraphic records, sometimes at centimeter scale stratigraphic resolution and $10^4$-year scale temporal resolution. The encoding of the periodic signal tracks changes in sediment (rock) accumulation rate and can be deconvolved through statistical analysis into robust durations of time, a strength that makes astrochronology an ideal tool for fine-scale investigations of geologic proxy records. However, perhaps the biggest limitation of astrochronology is that, in the absence of independent constraints, it typically produces "floating" chronologies that lack definitive anchoring to numerical time scales.

Combining floating astrochronologies and radioisotopic dates into an integrated model of age is an attractive prospect, as it leverages the strengths and overcomes the limitations of both data sources. Here we present a freely available R package (`astroBayes`; *Bayesian Astrochronology*) for joint Bayesian inversion of astrochronologic records and radioisotopic dates to develop high-precision age-depth models for stratigraphic sections. Following introduction of the new method, we investigate the sensitivity of `astroBayes` age-depth model construction to a variety of geologic scenarios, including varying the number and stratigraphic position of radioisotopic dates and the presence or absence of depositional hiatuses. We also present a case study from the Bridge Creek Limestone Member (Greenhorn Formation) of the Western Interior Basin (Meyers et al., 2012), where we refine the age of the Cenomanian–Turonian boundary using `astroBayes`.

The `astroBayes` method has several strengths over existing "dates only" age-depth models (Blaauw and Christen, 2011; Trayler et al., 2020; Haslett and Parnell, 2008; Keller, 2018). The inclusion of astrochronological data allows more densely constrained sedimentation models which results in an overall reduction in model uncertainty. Furthermore, these age-depth models are anchored in numerical time while simultaneously preserving astrochronologic durations minimizing "tuning" assumptions and potential missassignment of Milankovitch frequencies. These properties make the joint inversion ideal for correlating individual proxy records to other global records, enhancing our ability to constrain phase relationships and mechanisms of Earth System evolution.

## 2    Theory

### 2.1    Astrochronology

Quasiperiodic variations in Earth's orbital and rotational parameters impact the spatial and temporal distribution of sunlight on the planet's surface, and thus have the potential to alter regional and global climate. Such quasiperiodic climate changes can influence sedimentation and be preserved in the geologic archive, providing a dating tool for developing astronomical timescales, or astrochronologies. The astronomical variations include orbital eccentricity, with modern periods of 0.405 Myr and ~0.1 Myr, axial tilt (obliquity) with a dominant period of ~0.041 Myr today, and axial precession (or more specifically, "climatic precession"), with multiple periods near ~0.02 Myr today (Laskar, 2020). Solar system chaos limits reliable calculation of the full theoretical eccentricity solution to ~50 Ma, although the 'long eccentricity' cycle of 0.405 Myr is the most stable and likely suitable for use throughout the Phanerozoic (Laskar, 2020). Recently, Hoang et al. (2021) presented a new probabilistic model that permits estimation of all eccentricity cycle periods and their uncertainties throughout Earth history. In addition to Solar system chaos, Earth's dynamical ellipticity and tidal dissipation influence the temporal evolution of the precession and obliquity cycle periods, making them shorter in the geologic past, and there exist models of varying complexity for their estimation (Berger et al., 1992; Laskar et al., 2004; Waltham, 2015; Farhat et al., 2022; Laskar, 2020). Additional sources of uncertainty in floating astrochronologies include: (1) contamination of the astronomical-climate signal by other climatic and sedimentary processes, (2) spatial distortion of the astronomical cycles in the stratigraphic record including hiatus, and (3) uncertainties in the temporal calibration/interpretation of the observed spatial rhythms (Meyers, 2019). The design of the `astroBayes` approach carefully considers these sources of uncertainty.

### 2.2    Radioisotope Geochronology

Radioisotope geochronology utilizes the radioactive decay of a long-lived parent isotope to its daughter product within a closed geologic system to the determine its age. Temporal information is quantified in the evolving ratio of daughter to parent, as a function of the decay constant(s) of the constitutive nuclear reactions. In the case of sedimentary strata in deep time, these geologic systems are either radioisotopes captured in rapidly erupted and deposited igneous mineral grains in discrete

interbedded volcanic tuff horizons (U-Pb in zircon or K-Ar [implemented as the $^{40}$Ar/$^{39}$Ar technique] in feldspar), or endogenous sediment-bound radioisotopes that are fractionated during depositional processes at the sediment-water interface (Re-Os in organic-matter-bearing sedimentary rocks). The details of application of high-precision radioisotopic dating in the stratigraphic record may be found in reviews by Bowring and Schmitz (2003), Jicha et al. (2016), and Schmitz et al. (2020). The age interpretation is generally the result of an ensemble of measured ratios and/or dates interpreted as a model age, for example a weighted mean of numerous single crystal dates (U-Pb and $^{40}$Ar/$^{39}$Ar), a Bayesian estimation of the eruption age from the variance of those single crystal dates (Keller et al., 2018), or an isochronous relationship between sample aliquots (Re-Os). Radioisotopic model ages have an uncertainty that is usually described by a Gaussian probability function. In the case of either volcanic tuffs or endogenous sedimentary dating, the age constraints come from a restricted number of specific sampling horizons, which are generally stochastically present, preserved, and/or sampled within a stratigraphic succession.

## 2.3  Bayesian Statistics

The Bayesian statistical approach aims to determine the most probable value of unknown parameters given *data* and *prior* information about those parameters. This is formalized in Bayes' equation:

$$P(parameters|data) \propto P(data|parameters) \times P(parameters) \qquad (1)$$

The first term on the righthand side of eq. 1, known as the likelihood, is the conditional probability of the data, given a set of model parameters. The second term represents any prior beliefs about these model parameters. The left-hand side is the posterior probability of the model parameters. Bayes' equation is often difficult or impossible to solve analytically, and instead the posterior distribution is evaluated using Markov Chain Monte Carlo methods (MCMC) to generate a representative sample, which assuming a properly tuned MCMC process (Haario et al., 2001), should have the same properties (mean, median, dispersion, etc.) as the theoretical posterior distribution (Gelman et al., 1996).

## 2.4  Bayesian Age-Depth Modeling

Existing Bayesian methods for age-depth model construction rely on sedimentation models that link stratigraphic position to age through mathematical functions that approximate a sedimentation process conditioned through dated horizons throughout a stratigraphic section, which are then used to estimate the age and uncertainty at undated points (Blaauw and Heegaard, 2012). A variety of Bayesian approaches have been proposed to construct age-depth models including `Bchron` (Haslett and Parnell, 2008), `rbacon` (Blaauw and Christen, 2011), and `Chron.jl` (Schoene et al., 2019; Keller, 2018). While these methods vary considerably in their mathematical and computational framework, most share two fundamental characteristics. First, they treat sediment accumulation as a stochastic process where accumulation rate is allowed to vary randomly and considerably throughout a stratigraphic section. Second, they use this stochastic sediment accumulation model in tandem with discrete point-estimate likelihoods of numerical age, usually in the form of radioisotopic dates (e.g., $^{40}$Ar/$^{39}$Ar, U-Pb, $^{14}$C), as the basis for chronology construction. This leads to "dates-only" chronologies with widely variable uncertainties (Trachsel and Telford, 2017; Telford et al., 2004; De Vleeschouwer and Parnell, 2014) that are largely a function of data density. That is, modeled

age errors are lower in areas where there are more point-estimate age determinations, and age errors are higher in areas with less data, leading to "sausage" shaped uncertainty envelopes (De Vleeschouwer and Parnell, 2014).

Previous Bayesian approaches for linking astrochronology and radioisotopic dates have taken numerous approaches, including: (1) solely focusing on improving the ages of radioisotopically dated horizons using astrochronology (Meyers et al., 2012); (2) relying on post-hoc comparisons of computed astrochronologic and radioisotopic durations to accept or reject accumulation

models in the Markov Chain Monte Carlo process (De Vleeschouwer and Parnell, 2014) or (3) "transforming" astrochronologic durations into age likelihoods via anchoring to other radioisotopically dated horizons (Harrigan et al., 2021). Meyers et al. (2012) modified the Bayesian "stacked bed" algorithm of Buck et al. (1991) to incorporate known astrochronologic durations between dated horizons, allowing for the improvement of Cretaceous radioisotopic age estimates using astrochronology, and the age of the Cenomanian/Turonian boundary. Their approach, however, did not explicitly model posterior age estimates for

intervening strata in the Bayesian inversion. De Vleeschouwer and Parnell (2014) recalibrated the Devonian time scale and calculated new stage boundaries using a two-step process. First the authors generated a continuous Bayesian age-depth model using the `Bchron` R package (Haslett and Parnell, 2008) and then performed a post-hoc rejection of model iterations that violated previously derived astrochronologic stage durations. While these results are consistent with both data types, the two-step process does not fully integrate and leverage astrochronology in the age-model construction. Harrigan et al. (2021) further

refined the Devonian timescale by using a modified version of `Bchron` (Trayler et al., 2020). The authors used a Monte Carlo approach to convert astrochronology derived durations into stage boundary ages which were then included as inputs alongside radioisotopic dates for Bayesian modeling. Each of these methods requires external processing and interpretation of astrochronologic data, either to derive durations or to transform them into a form (i.e., age ± uncertainty) that is amenable to inclusion within existing models. In this study we present a new approach designated `astroBayes`, which fully leverages

the advantages of radioisotopic ages and astrochronology by explicitly including both in the Bayesian inversion.

# 3 Methods

## 3.1 Model Construction

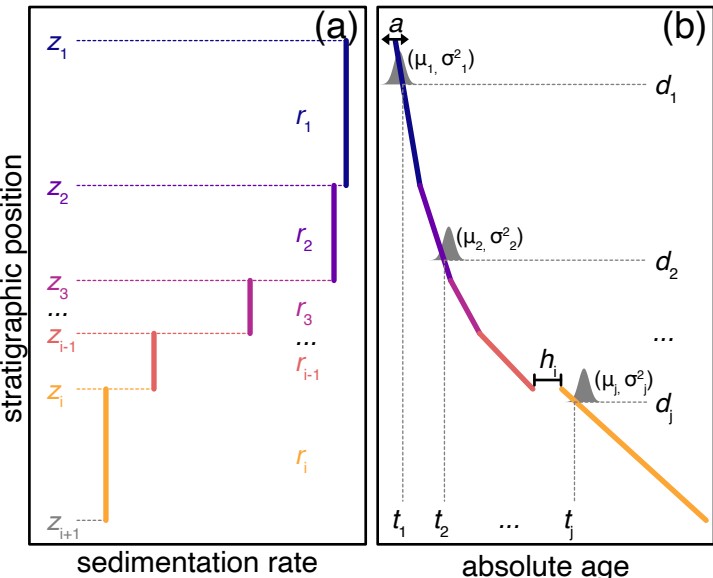

**Figure 1: Schematic of model parameters. A) A simple five-layer sedimentation model. B) The sedimentation model from panel A)**
150 **transformed and anchored as an age-depth model. See Table 1 for an explanation of each parameter.**

**Table 1: Summary of model parameters.**

| Parameter | Explanation |
|---|---|
| $r$ | sedimentation rate (m/Ma) |
| $z$ | layer boundary positions (stratigraphic positions) |
| $a$ | anchoring age (Ma) |
| $D, d$ | depth (stratigraphic positions; transformation of $z$) |
| $h$ | hiatus duration (Ma) |
| $T, t$ | age (Ma; transformation of $r$ and $z$) |
| $f$ | orbital target frequencies (cycles/Ma) |
| *data* | astrochronologic data (value vs stratigraphic position) |
| *dates* | radioisotopic dates (Ma) |

The inputs for `astroBayes` consists of measurements of a cyclostratigraphic record (*data*) (e.g., $\delta^{18}O$, XRF scans, core resistivity, etc.), and a set of radioisotopic dates (*dates*) that share a common stratigraphic scale. The user also specifies a set of appropriate target frequencies (*f*; eccentricity, obliquity, precession) for use in probability calculations. Developing an age-
155 depth model from these records requires 1) a likelihood function that reflects the probability of both data types; 2) a common set of model parameters to be estimated; and 3) in the case of continuous age-depth modeling, a model that reflects the best

approximation of sediment accumulation. We focus on estimating the probability of sedimentation rate as the basis for the `astroBayes` age-depth model. Since sedimentation rate is expressed as depth-per-time (e.g., m/Myr, cm/kyr) it directly links stratigraphic position to relative age to create floating age models, and when combined with radioisotopic dates, generates models anchored in numerical time.

Existing Bayesian age-depth modeling approaches approximate sedimentation as a relatively large number of piecewise linear segments. Sedimentation rate can vary substantially between segments, leading to the "sausage-shaped" uncertainty envelopes that characterize these models (Trachsel and Telford, 2017; De Vleeschouwer and Parnell, 2014; Parnell et al., 2011). However, this model of sedimentation is not ideal for the construction of astrochronologies because fluctuations in sedimentation rate can be constrained by preserved astronomical frequencies as spatial stretching or compression of the preserved rhythm. As our nominal approach, we adopt a sedimentation model with a small number (< 10) of layers of consistent sedimentation rate, following a common astrochronologic approach of minimizing fine-scale adjustments to sedimentation rate (Muller and MacDonald, 2002; Malinverno et al., 2010). However, the general approach can be adapted to include any number of layers.

Malinverno et al. (2010) presented a simple sedimentation model appropriate for astronomical tuning of sedimentary records and we use their framework as the starting basis for the joint inversion. The sedimentation model consists of two sets of parameters. The first is a vector of sedimentation rates ($r$), and stratigraphic boundary positions ($z$) that define regions ("layers") of constant sedimentation (Fig. 1A). For example, the model shown in Fig. 1A has 11 parameters, five sedimentation rates ($r_1 - r_i$) and six layer-boundaries ($z_1 - z_i$). This model formulation allows step changes in sedimentation rate at layer boundaries ($z$) but otherwise holds sedimentation rate ($r$) constant within each layer.

The selection of layer boundary-positions is an important user defined step, that is informed by detailed investigation of the cyclostratigraphic data. Evolutive harmonic analysis (EHA) is a time-frequency method that can identify changes in accumulation rate by tracking the apparent spatial drift of astronomical frequencies. Expressed as cycles/depth, high amplitude cycles may "drift" towards higher or lower spatial frequencies throughout the stratigraphic record. Assuming these spatial frequencies reflect relatively stable astronomical periodicities, the most likely explanation of those spatial shifts is therefore stratigraphic changes in sedimentation rate (Meyers et al., 2001). That is, stability in spatial frequencies reflects stability in sedimentation rate, allowing sedimentation to be approximated by a small number of piecewise linear segments.

We visually inspected EHA plots to develop simple sedimentation models (e.g., Fig. 1B) for our testing data sets. We choose layer boundary-positions ($z_1 - z_i$) by identifying regions with relatively stable spatial frequencies (see Fig. 2). For example, in Fig. 2C, there is a continuous high-amplitude frequency-track between 2-4 cycles/m. Based on visual shifts in this frequency, we choose three layer-boundaries, such that this frequency track can be approximated by a vertical line within each layer. In the computation, we also allow the layer boundary-positions to vary randomly (within a user specified stratigraphic range) to account for stratigraphic uncertainties in boundary-positions that arise from the fidelity and our inspection of the of the data, similar to the Bayesian cyclostratigraphic approach of Malinverno et al. (2010).

Together $r$ and $z$ can also be transformed to create an age-depth model consisting of piecewise linear segments that form a floating age-depth model (Fig. 1B). This floating model can be anchored in numerical time by adding a constant age ($a$) to the

floating model at every stratigraphic position. Optionally, sedimentary hiatuses can also be included in the model in a similar manner by adding the duration of a hiatus ($h$) at any of the layer boundary positions to all of the points below the stratigraphic position of the hiatus.

## 3.2    Probability Estimation

Together the vectors of sedimentation rates ($r$), layer boundaries ($z$), and anchoring age ($a$) can be used to calculate an anchored *age-depth model* that consists of a series of piecewise linear segments (Fig. 1B). The slope (m/Ma) and length of these segments is controlled by the sedimentation rates ($r$) and layer boundary positions ($z$), while the numerical age is controlled by the anchoring constant ($a$). Hiatuses ($h$) at each layer boundary can offset the age-depth model in time. The anchored age-depth model now consists of a vector of stratigraphic positions ($D$) and a corresponding vector of ages ($T$) that relate

stratigraphic position to numerical age. The probability of this age-depth model can be assessed by calculating the probability of the sedimentation rates ($r$) and anchoring constant ($a$) given an astrochronologic record (*data*) and a series of radioisotopic dates (*dates*).

### 3.2.1    Probability of an Astronomical Model

We follow the approach of Malinverno et al. (2010) to calculate the probability of our data given a sedimentation rate and set

of target astronomical frequencies (*f*).

$$P(data|r, f) \propto exp\left[\frac{C_{data}(f)}{C_{background}(f)}\right] \qquad (2)$$

Where the data is the astrochronologic record, $r$ is a sedimentation rate, and $f$ is an astronomical frequency (e.g., Table 2), $C_{data}$ is the periodogram of the data, and $C_{background}$ is the red noise background. The probability in eq. 2 is calculated independently for each model layer (i.e., between adjacent $z$'s), and the overall probability is therefore the joint probability of all layers. eq. 2

calculates the concentration of spectral power at specified astronomical frequencies, where a given sedimentation rate is more probable if it causes peaks in spectral power that rise above the red noise background to "line up" with astronomical frequencies. The red noise background is approximated using a lag-1 autoregressive process (AR(1); Gilman et al. (1963)) which provides a useful stochastic model for climate and cyclostratigraphy (Gilman et al., 1963; Hasselman, 1976).

**Table 2: Astronomical frequencies used for model testing and validation for the two synthetic testing data sets (discussed below).**
**The precession and obliquity terms are based on the LA04 solution (Laskar et al., 2004), and the eccentricity terms are based on the LA10d solution (Laskar et al., 2011).**

| Period (Ma) | Frequency (1/Ma) | Cycle |
| --- | --- | --- |
| 0.4057 | 2.4650 | eccentricity |
| 0.1307 | 7.6500 | eccentricity |
| 0.1238 | 8.0750 | eccentricity |
| 0.0989 | 10.1150 | eccentricity |

| Period (Ma) | Frequency (1/Ma) | Cycle |
|:-----------:|:----------------:|:-----:|
| 0.0949 | 10.5400 | eccentricity |
| 0.0410 | 24.4100 | obliquity |
| 0.0236 | 42.3358 | precession |
| 0.0223 | 44.8055 | precession |
| 0.0190 | 52.6497 | precession |
| 0.0191 | 52.4448 | precession |

### 3.2.2 Probability of Radioisotopic Dates

The anchored age-depth model now consists of two paired vectors that relate stratigraphic position ($D$) to numerical time ($T$). The stratigraphic positions of the radioisotopic dates $[d_1 \ldots d_j]$ and their corresponding ages $[t_1 \ldots t_j]$ are a subset of $D$ and $T$, respectively. We therefore define the probability of the modeled age ($T$) at a depth ($D$), given a set of dates as:

$$P(T|dates) = \prod_{j=1}^{n} N\left(\mu_j, \sigma_j^2\right) \qquad (3)$$

Where $N$ is a normal distribution with a mean ($\mu$) and variance ($\sigma^2$). $\mu_j$ is the weighted mean age and $\sigma^2_j$ is the variance of the $j^{th}$ radioisotopic date at stratigraphic position $d_j$. Notice that while $d$ and $t$ are continuous over the entire stratigraphic section, only the stratigraphic positions that contain radioisotopic dates influence the probability of the age model. In effect, this probability calculation reflects how well the age model "overlaps" the radioisotope dates, where modeled ages that are closer to the radioisotopic dates are more probable (Fig. 1B (Schoene et al., 2019; Keller, 2018).

### 3.2.3 Overall Probability and Implementation

The overall likelihood function of an anchored age-depth model is now the joint probability of eq. 2 and eq. 3. We use a vague uniform prior distribution where sedimentation rate may take any value between a specified minimum and maximum value. `astroBayes` estimates the most probable values of sedimentation rate, anchoring age, and hiatus duration(s) using a Metropolis-Hasting algorithm and an adaptive Markov Chain Monte Carlo (MCMC) sampler (Haario et al., 2001) to generate a representative posterior sample for each parameter. The complete model is available as an R package called `astroBayes` (*Bayesian astrochronology*) at github.com/robintrayler/astroBayes.

## 3.3 Testing and Validation

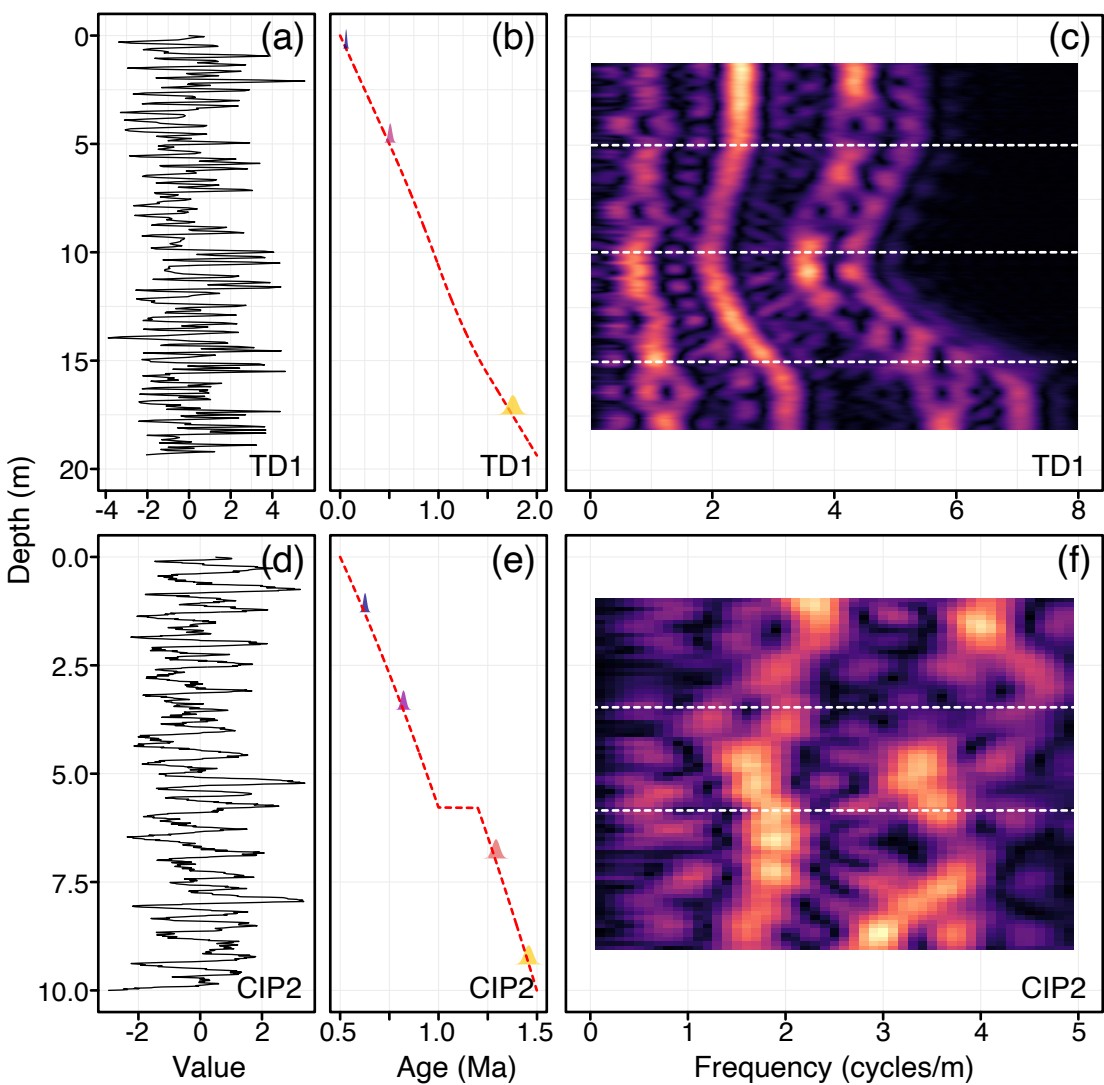

**Figure 2: Synthetic testing data sets used for model validation. A, D) The synthetic cyclostratigraphic records for TD1 and CIP2. B, E) True age-depth models for both data sets. The colored probability distributions are the synthetic radioisotopic dates used for model stability testing (Table 3). C, F) Evolutive harmonic analysis of panels A (3m window size, 0.1m step size using 3-2π prolate tapers) and D (2m window size, 0.1m step size, using 3-2π prolate tapers). Lighter colors indicate higher spectral amplitude. The horizontal dashed lines are layer boundary positions ($z$) chosen by visual inspection of the evolutive harmonic analysis results.**

We tested `astroBayes` using two synthetic data sets that consist of a known age-depth model and a paired cyclostratigraphic record. The first dataset (TD1) consists of a simple sedimentation model that was used as an earth system transfer function to distort a normalized eccentricity-tilt-precession (ETP; Laskar et al. (2004)) time series (with equal contribution of each astronomical parameter) to generate a synthetic cyclostratigraphic record (Fig. 2A). This 2 million year ETP signal was

translated into a stratigraphic signal using a stable sedimentation rate of 7.5 m/Myr for the first 0.500 Myr (the oldest portion of the record), followed by a linear sedimentation rate increase to 12.5 m/Myr until 1.0 Myr, then a linear sedimentation rate decrease to 10 m/Myr until 1.5 Myr, and finally a stable sedimentation rate of 10 m/Myr for the youngest stratigraphic interval. The second dataset (CIP2) was originally published by Sinnesael et al. (2019) as a testing exercise for the Cyclostratigraphy Intercomparison Project which assessed the robustness and reproducibility of different cyclostratigraphic methods. The CIP2 dataset was designed to mimic a Pleistocene proxy record with multiple complications including nonlinear cyclical patterns and a substantial hiatus. For full details on the construction of the CIP2 dataset see Sinnesael et al. (2019) and cyclostratigraphy.org. For each of our testing schemes, outlined below, we used the true age-depth model to generate synthetic radioisotopic dates (with uncertainties) from varying stratigraphic positions. The combination of synthetic cyclostratigraphic data and simulated radioisotopic dates form our synthetic model inputs.

We assessed model performance using two metrics. First, we assessed model accuracy and precision by calculating the proportion of the true age-depth model that fell within the 95% credible interval (95% CI) of our model posterior. We assume that a well-performing model should contain the true age model in most cases. This method has been used previously to assess performance of existing Bayesian age-depth models (Parnell et al., 2011; Haslett and Parnell, 2008). Second we monitored the variability of the model median (50%) and lower and upper bounds (2.5% and 97.5%) of the credible interval.

### 3.3.1 Reproducibility and Stability

**Table 3: Dates used as inputs for reproducibility & stability testing of the synthetic test cases (TD1 and CIP2).**

| Data Set | Sample | Age$\pm1\sigma$ (Ma) | Position (m) |
|----------|--------|---------------------|--------------|
| TD1 | A | 0.069±0.01 | 0.64 |
| | B | 0.520±0.02 | 5.17 |
| | C | 1.790±0.05 | 17.48 |
| CIP2 | D | 0.062±0.009 | 1.24 |
| | E | 0.820±0.012 | 3.49 |
| | F | 1.290±0.019 | 6.99 |
| | G | 1.460±0.022 | 9.49 |

To assess the reproducibility and stability of `astroBayes` we generated 1,000 individual age-depth model Bayesian inversions for each synthetic testing dataset to assess model reproducibility and stability. We used the same input data for the Bayesian inversions: the same cyclostratigraphic records (Fig. 2), astronomical frequencies (Table 3) and radioisotopic dates (Table 3). Each simulation ran for 10,000 MCMC iterations to allow sufficient exploration of parameter space and posterior convergence to the target stationary distribution. The adaptive Metropolis-Hastings proposal algorithm adequately stabilized each Markov Chain after an initial discarded "burn-in" period of 1,000 iterations.

### 3.3.2 Sensitivity Testing with the Synthetic Models

We tested the sensitivity of our age-depth model results to both the number and stratigraphic position of radioisotopic dates. We randomly generated a set of dates from the underlying sedimentation model using Monte Carlo methods. The uncertainty ($1\sigma$) was set at 1.5% of the age. These dates and uncertainties were used as radioisotopic age likelihoods along with the synthetic astrochronologic records. We repeated this procedure 1,000 times using 2, 4, 6, or 8 dates for a total of 4,000 simulations per testing data set (i.e., 4,000 for CIP2 and TD1). Each simulation ran for 10,000 MCMC iterations with a 1,000 iteration "burn-in".

Since the CIP2 data set includes a significant hiatus (Sinnesael et al., 2019) we also investigated the influence of the number and stratigraphic position of radioisotopic dates on the quantification of the hiatus duration. Estimating hiatus duration requires at least one date above and below the stratigraphic position of a hiatus. Consequently, we added an additional constraint when generating synthetic dates from the CIP2 dataset to ensure that the hiatus was always bracketed by at least two dates. For each of the sensitivity validation models (2, 4, 6, and 8 dates) we benchmarked the stratigraphic distance between the hiatus and the nearest date.

### 3.3.3 Sensitivity to Outlier Ages

We also tested the sensitivity of `astroBayes` to the inclusion of outlier ages. We repeated the tests from Section 3.3.2, with one additional step. After the generation of stratigraphically-randomly distributed dates, we used Monte Carlo methods to select one date from each testing data set. This date was then randomly adjusted by $\pm1\sigma$ to $\pm4\sigma$. This creates a date that is either broadly comparable with the underlying true age model (e.g., $\pm1\sigma$ to $\pm2\sigma$), or outlier ages that may introduce stratigraphic miss-matches (e.g., $\pm3\sigma$ to $\pm4\sigma$). We choose to introduce these more subtle outliers, since we feel more extreme outlier ages can often be identified and excluded *a priori* based on inspection of the radioisotopic data (Michel et al., 2016). We repeated this procedure 1,000 times using either 2, 4, 6, or 8 dates for each data set (as in the section above), so that 1/2, 1/4, 1/6, and 1/8 dates would be considered an outlier. Each simulation ran for 10,000 MCMC iterations with a 1,000 iteration "burn-in".

# 4 Results

## 4.1 Model Validation

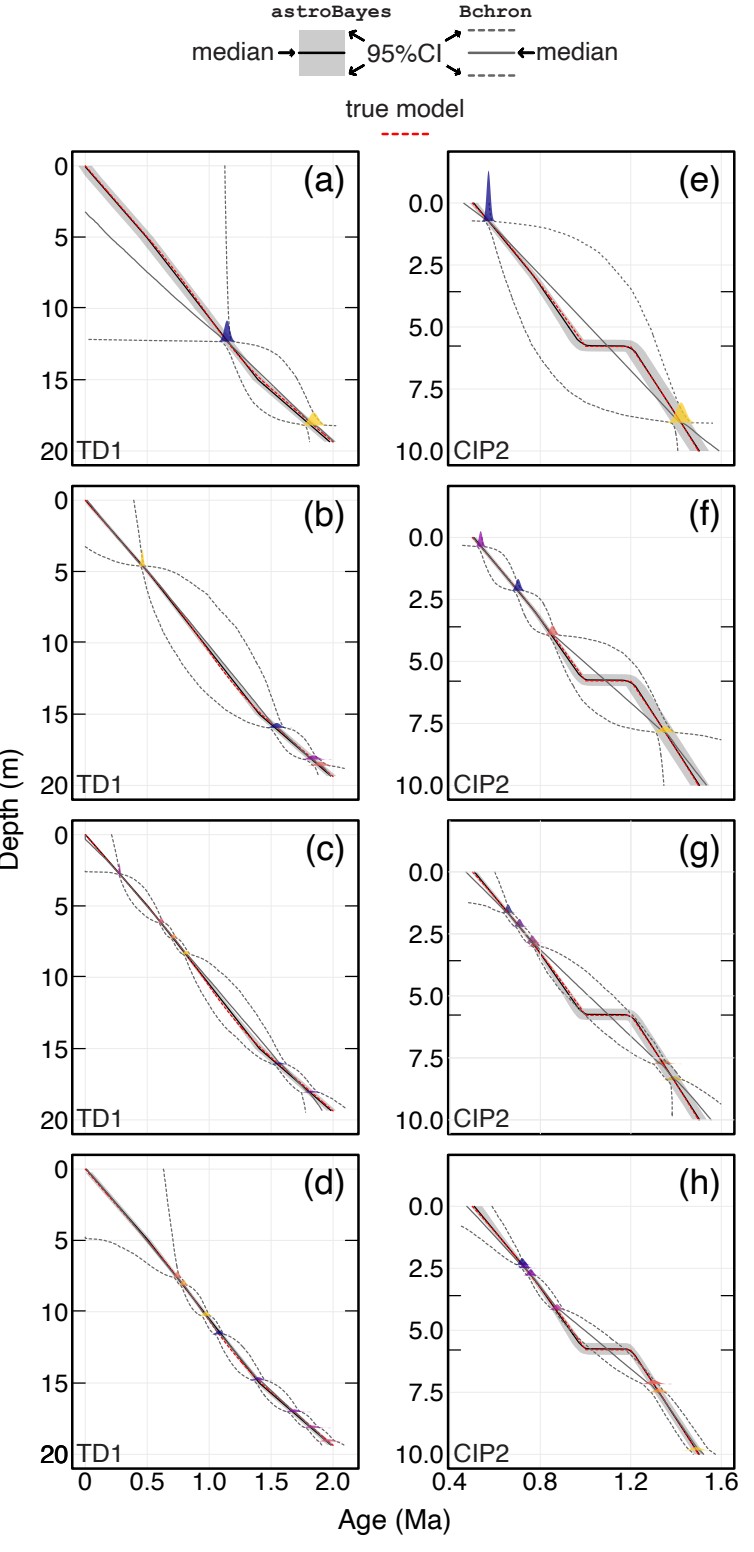

**Figure 3: Example age-depth models of the synthetic TD1 and CIP2 test data sets with randomly placed dates shown as colored Gaussian distributions. Interior tick marks on the vertical axis of each panel indicate the layer boundary positions (see also the horizontal dashed lines in Fig. 2C and F). The dates were randomly generated from the true age-depth model (dashed red line). The black line and shaded grey region are the `astroBayes` model median and 95% credible interval. The dark grey solid and dashed lines are `Bchron` models generated using only the radioisotopic dates as model inputs. Panels A - D) 2, 4, 6, and 8 date models for the TD1 synthetic data. Panels E - H) 2, 4, 6, and 8 date models for the CIP2 synthetic data. Note that the left and right columns have different vertical and horizontal scales.**

Reproducibility tests indicate that the `astroBayes` model converges quickly and its parameter estimates remain stable across model runs. Individual trace plots for each parameter (sedimentation rates, anchor age, hiatus duration [CIP2 only]) for the TD1 and CIP2 synthetic data sets stabilized quickly and appear visually well-mixed indicating adequate exploration of the parameter space (see supplemental figures Figs. S1 - S4). Similarly, posterior kernel density estimates of each parameter were indistinguishable among the 1,000 simulations. The model median and 95% credible interval were likewise stable and varied by no more than ±0.005 Myr (2σ) for both testing data sets.

Model accuracy does not appear to be particularly sensitive to the number or stratigraphic position of dates as the true age-depth model fell within the 95% credible interval of the `astroBayes` posterior 99% of the time with no clear bias towards greater or fewer dates (Fig. 3). For the CIP2 data set, other than the requirement that there is at least one date above and below the hiatus, the stratigraphic position of the dates does not appear to have a strong influence on hiatus quantification and in all cases the true hiatus duration (0.203 Ma) was contained within the 95% CI of the hiatus duration posterior (*h*; Fig. 4). `astroBayes` is somewhat sensitive to the inclusion of subtle outlier radioisotopic dates. The inclusion of outlier ages lowered the proportion of the true age-depth models that fell within the 95% credible interval of the `astroBayes` to 89% for TD1, and 88% for CIP2. The relative percentage of outlier ages also does not appear to have a strong influence.

The number of radioisotopic dates appears to have the strongest effect on overall model uncertainty (see also: Blaauw et al. (2018)). As the number of dates increase the width of the 95% credible interval shrinks and approaches the input uncertainty of the radioisotopic dates (Fig. 3). Crucially however, the uncertainties never "balloon" (e.g., compare `astroBayes` with `Bchron` results in Fig. 3) and are usually close to the uncertainty of the dates, unlike "dates-only" age-depth models (De Vleeschouwer and Parnell, 2014).

## 5    Discussion

### 5.1    Developing Sedimentation Models and Constraining Uncertainty

Clearly, our choice of a simple sedimentation model for Bayesian inversion influences age-depth model construction. Since eq. 2 is calculated layer-by-layer, a limitation of our model is that each layer must contain enough time and astrochronologic data to resolve the astronomical frequencies (*f*) of interest. Both the astrochronologic data and radioisotopic dates can inform sedimentation model construction. First, the radioisotopic dates can be used to calculate average sedimentation rates which to a first approximation can then inform the length of sedimentation model layers needed to capture specific astronomical cycles

(e.g., eccentricity). For example, Table 3 contains the dates and stratigraphic positions used for inputs for TD1 stability testing (see Section 3.3.1). A time difference of 1.72 Myr between the uppermost and lowermost dates separated by 16.84 meters implies an average sedimentation rate of ~9.8 m/Myr or alternatively ~0.1 Ma/m. A sedimentation model with a layer thickness of 1 meter would not reliably resolve long (~0.405 Ma) and short (~0.1 Ma) eccentricity cycles and would only weakly resolve obliquity (~0.41 Ma) and precession scale cycles (~0.02 Ma) within each layer. The choice of layer thickness is therefore dependent on both the average sedimentation rate, the cyclostratigraphic sampling rate, and the dominant astronomical signals present in the data. Records dominated by eccentricity scale fluctuations will necessarily require layer thicknesses that capture longer timescales than records dominated by higher frequency obliquity and precession scale variations. Future model development could semi-automate much of this starting model construction, optimizing the number and length of layers. However, a critical prerequisite is that the cyclostratigraphic data series has a sampling rate sufficient to reliably capture the highest frequency of interest (e.g., precession).

A potential criticism of our approach is that our choice of a simple Bayesian sedimentation model artificially reduces overall model uncertainties. Since we do not allow sedimentation rate to vary randomly at all points throughout the stratigraphy, our model avoids the inflated ("ballooning") credible intervals that characterize "dates-only" age-depth models (i.e., `Bchron`, `rbacon`, `Chron.jl`). Indeed, Haslett and Parnell (2008) consider this minimum assumption of smoothness as a fundamental feature of age-depth modeling as there is *"no reason a priori to exclude either almost flat or very steep sections"*. Although Blaauw and Christen (2011) consider some smoothness desirable, both modeling approaches allow sedimentation rate to vary randomly and considerably in the absence of other constraints. However, we feel that astrochronology provides a clear, strong constraint on the stratigraphic variability in sedimentation rate. Astronomical tuning approaches show that changes in sedimentation rate can be unrelated to astronomical forcing yet be preserved in the spatial representation of the astronomical cycles (Muller and MacDonald, 2002; Malinverno et al., 2010) and stratigraphic investigation of preserved astronomical frequencies often reveals long periods of near constant sedimentation rates (Shen et al., 2022; Sinnesael et al., 2019; Meyers et al., 2001). Therefore, the addition of cyclostratigraphic data to age-depth model construction allows for the informed development of simpler sedimentation models which result in substantially lower uncertainties.

## 5.2    Hiatus Duration Estimation

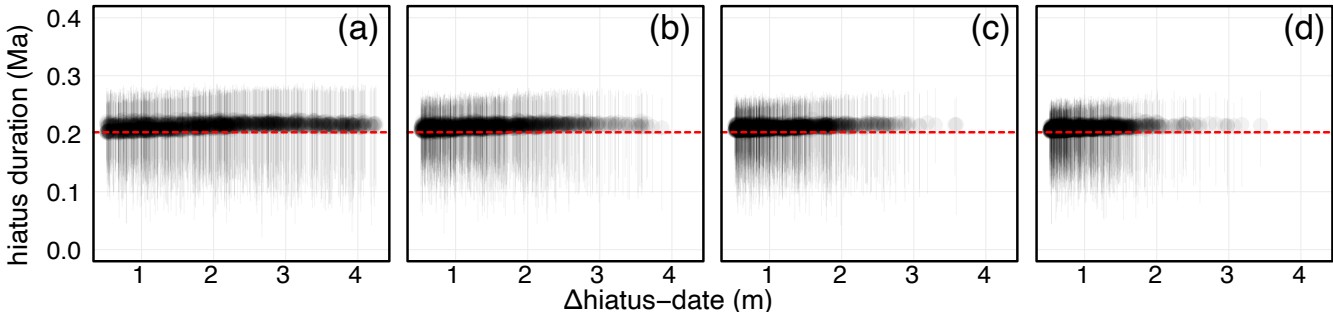

**Figure 4: Hiatus duration versus the stratigraphic distance between the hiatus and the nearest radioisotope date for the CIP2 data set. The points are the model median, and the error bars are the 95% credible interval. The red line is the true hiatus duration of 0.203 Ma. A-D) Models with 2, 4, 6, and 8 ages respectively.**

The ability to estimate hiatus durations is a significant strength of the atroBayes modeling framework. Hiatuses in stratigraphic records significantly complicate the interpretation of biologic and geochemical proxy records. Detecting and resolving the duration of hiatuses is therefore important to ensuring the accuracy of age-depth models. In principle, hiatuses can be detected and quantified from cyclostratigraphic records alone (Meyers and Sageman, 2004; Meyers, 2019). However, these approaches can be skewed towards minimum hiatus duration and are sensitive to distortions of the astronomical signal from other non-

hiatus sources (Meyers and Sageman, 2004). `astroBayes` relies on both astrochronology and radioisotopic geochronology to estimate the duration of one or more hiatuses with the joint inversion of astrochronology and radioisotopic ages controlling the sedimentation rates (slopes) above and below them, while also determining the absolute ages above and below hiatuses. However, it should be noted that there are two potential weaknesses of this approach to estimating hiatus duration. First, since hiatus positions are user defined, the stratigraphic position of a hiatus must be known *a priori* and must be informed by geologic

(i.e., a visible unconformity) or cyclostratigraphic data (Meyers and Sageman, 2004). In both the CIP2 testing data set and the Bridge Creek Limestone Member case study (discussed below), the stratigraphic positions of the hiatuses were known in advance. The second weakness is that `astroBayes` cannot reliably estimate durations for hiatuses unconstrained by radioisotopic dates. If a hiatus only has radioisotopic dates stratigraphically above or below, the undated side is unconstrained and duration estimates tend to wander towards an infinite duration. Likewise, if a model layer is bounded by two hiatuses and

the layer does not contain any radioisotopic dates, then `astroBayes` cannot reliably resolve the duration of the bounding hiatuses and will tend to "split the difference". However, when hiatuses are well-constrained by radioisotopic dates, `astroBayes` allows the estimation of robust uncertainties of hiatus duration and is a powerful tool when there is external sedimentological or astronomical evidence for hiatuses, as shown in the Bridge Creek Limestone Member case study below.

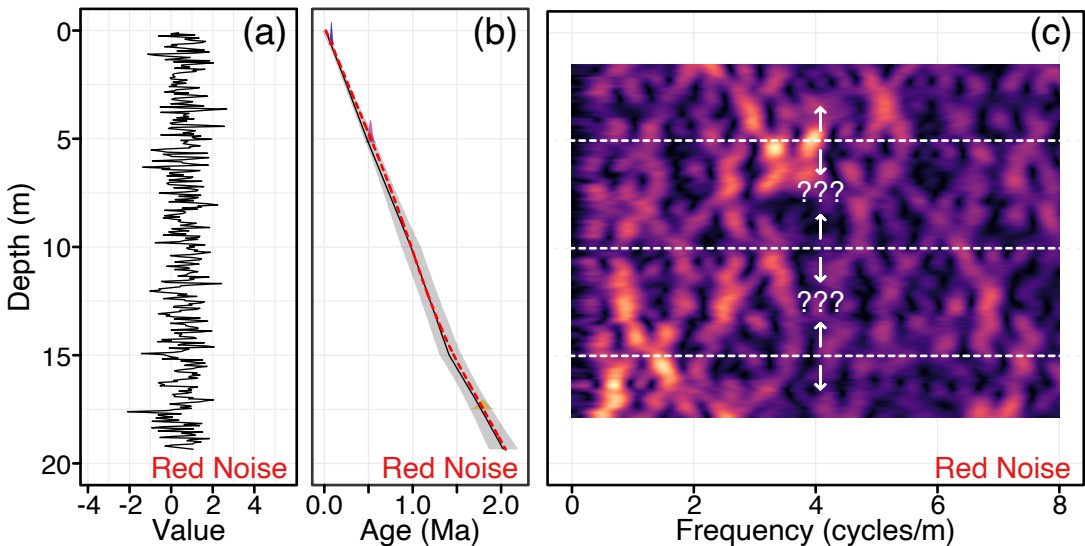

**Figure 5: Results of `astroBayes` modeling of the TD1 testing dataset, with the cyclostratigraphic data replaced by randomly generated AR(1) red-noise. A) Randomly generated AR(1) red-noise B) Age-depth model generated using the correct dates, frequencies, and layer boundaries, and the red-noise cyclostratigraphic data C) Evolutive harmonic analysis of A) (3m window size, 0.1m step size using 3-2π prolate tapers). The dashed lines indicate the layer boundary positions used for other model testing (see Fig. 2). The arrows indicate the uncertainty in layer boundary position reflecting the lack of any stratigraphically stable and continuous frequencies in the data.**

Because `astroBayes` is available as an R package, it is straightforward to install and use, assuming familiarity with the R programming language (R Core Team, 2023). Given this, we feel we should discuss appropriate and inappropriate use of the modeling framework. First, `astroBayes` is not a method to test for the presence of statistically-significant astronomical signals and it does not include any null-hypothesis tests. There are a variety of statistical methods available to test for the presence of astronomical signals in the rock record (Huybers and Wunsch, 2005; Meyers and Sageman, 2007; Zeeden et al., 2015; Meyers, 2019) which should be used prior to `astroBayes` modeling. Instead, `astroBayes` is intended to be used to develop age-depth models after the presence of astronomical signals has been established using other methods. Similarly, `astroBayes` does not include automated outlier rejection for radioisotopic dates (Bronk Ramsey, 2009) and these data should be pre-screened following best practices for high precision geochronology (Michel et al., 2016; Schmitz and Kuiper, 2013).

`astroBayes` is software, and it is quite possible to generate an age-depth model from data that lacks any astronomical signals or contains outlier radioisotopic dates. Therefore, `astroBayes` makes three assumptions about the input data. 1) the cyclostratigraphic *data* has been vetted and has been shown to contain statistically significant astronomical signals using other astrochronologic testing approaches. 2) The user-specified layer boundary positions ($z$) have been informed by either careful inspection of the cyclostratigraphic *data* (e.g., time-frequency analysis such as EHA), and other geologic data (e.g., visible

facies changes), or both. 3) The radioisotopic dates have been prescreened and do not contain obvious outlier dates or violations of fundamental geologic principles (e.g., superposition).

For a simple example of an inappropriate use of `astroBayes`, we replaced the cyclostratigraphic *data* in the TD1 data set with randomly generated AR(1) red-noise. All other parameters (dates, layer boundaries, target frequencies) remained the same (see: Fig. 2, Table 3 and Table 2). Together, we used these data to generate an `astroBayes` age-depth model, shown in Fig. 5. The resulting age-depth model (Fig. 5B) looks superficially similar to the example models shown in Fig. 3. Since the radioisotopic dates still offer some limits on sedimentation rate, the median model still appears similar to the true age model. While the model credible interval is somewhat wider, it does not "balloon" and the overall uncertainties remain low compared to dates-only models (e.g., `BChron`). However, while this age-depth model looks superficially promising, it violates two of the assumptions discussed above. First, the "cyclostratigraphic" *data* (red-noise) does not contain any statistically significant astronomical periods, leading to meaningless probability calculations. Second, because the "cyclostratigraphic" *data* is random, it cannot be used to inform the placement of layer boundaries. Indeed the evolutive harmonic analysis shown in Fig. 5C shows no stratigraphically stable frequencies, making the layer boundary positions used for this example arbitrary and incorrect. The `astroBayes` modeling framework explicitly assumes a piecewise linear sedimentation model (Fig. 1) where sedimentation rate only varies at layer boundaries but is otherwise stable. Since for this example the "cyclostratigrapy" contains no astronomical signals, and the layer boundary positions cannot be reliably determined, `astroBayes` would be an inappropriate modeling tool.

## 5.4    Case Study: The Cenomanian-Turonian Bridge Creek Limestone Member

The Bridge Creek Limestone Member is the uppermost member of the Greenhorn Formation of central Colorado. It is primarily composed of hemipelagic marlstone and limestone couplets that extend laterally for over 1,000 km in the Western Interior Basin (Elder et al., 1994). These couplets are characterized by alternations from darker organic carbon-rich laminated clay and mudstones to lighter carbonate-rich, organic carbon-poor limestone facies. Previous work has reported Milankovitch scale cyclicity in the Bridge Creek Limestone Member through the application of statistical astrochronologic testing methods (Sageman et al., 1997, 1998; Meyers et al., 2001, 2012, 2008). Using U-Pb and $^{40}Ar/^{39}Ar$ ages from several bentonites throughout the section to provide temporal anchoring of the astrochronology, Meyers et al. (2012) previously calibrated the age of the Cenomanian-Turonian boundary as 93.90±0.15Ma (mean±95%CI) using an adaptation of the Bayesian "stacked bed" algorithm (Buck et al., 1991) that respects both stratigraphic superposition and astrochronologic durations between the dates and the boundary position. That work used the floating astrochronology of Meyers et al. (2001), based on analysis of a high stratigraphic resolution optical densitometry record (i.e., grayscale) of the Bridge Creek Limestone Member. Meyers and Sageman (2004) later quantified a brief hiatus in the Bridge Creek Limestone Member near the base of the *Neocardioceras juddii* ammonite biozone, at the top of limestone marker bed LS5 (Elder et al., 1994), with an estimated minimum duration of

0.079 – 0.0254 Ma. Sedimentologic evidence for the hiatus incudes the presence of a calcarenite cap at the top of LS5 at the basin center Pueblo, Colorado section (Meyers and Sageman, 2004).

**Table 4: Astronomical target periods used for the Bridge Creek Limestone Member `astroBayes` analysis. The precession and obliquity terms are based on the reconstruction of Waltham (2015) at 94 Ma, and the eccentricity terms are based on the LA10d solution (Laskar et al., 2011) from 0-20 Ma. We used the average of the two ~0.02 Myr and two ~0.018 Myr precession terms.**

| Period (Myr) | Frequency (1/Myr) | Cycle |
|---|---|---|
| 0.4057 | 2.46500 | eccentricity |
| 0.0940 | 10.54000 | eccentricity |
| 0.0989 | 10.11500 | eccentricity |
| 0.0504 | 19.82420 | obliquity |
| 0.0391 | 25.57545 | obliquity |
| 0.0279 | 35.82561 | obliquity |
| 0.0224 | 44.62294 | precession |
| 0.0186 | 53.74899 | precession |

We used `astroBayes` to develop two new age-depth models for the Bridge Creek Limestone Member using the the grayscale record of Meyers et al. (2001), a suite of target astronomical frequencies (Table 4), and two sets of radioisotopic dates, resulting in two alternative models. For the first model (*Meyers* model) we used the $^{40}Ar/^{39}Ar$ bentonite ages of Meyers et al. (2012), and for the second (*Updated* model) we used the updated $^{40}Ar/^{39}Ar$ ages of Jones et al. (2021) and Jicha et al. (2016). Note that since the A-bentonite has not been reanalyzed, both models use the Meyers et al. (2012) age for this sample (Table 5). We divided the Bridge Creek Limestone member grayscale record (Fig. 6A) into three layers based on the observed shifts in the high spectral amplitude frequency-track (~1.1 cycles/m) delineated about 6.7 meters height and at the reported hiatus at 2.7 meters height (Meyers and Sageman, 2004) depth (Fig. 6B).

**Table 5: Radioisotopic dates used used as model inputs for the two Bridge Creek Limestone Member age-depth models shown in Fig. 6.**

| Age Model | Sample | Age±1σ (Ma) | Position (m) | Source |
|---|---|---|---|---|
| Meyers | A-bentonite | 94.20±0.140 | 1.62 | Meyers et al. (2012) |
| | B-bentonite | 94.10±0.135 | 3.30 | Meyers et al. (2012) |
| | C-bentonite | 93.79±0.130 | 5.95 | Meyers et al. (2012) |
| | D-bentonite | 93.67±0.155 | 6.98 | Meyers et al. (2012) |
| Updated | A-bentonite | 94.20±0.140 | 1.62 | Meyers et al. (2012) |
| | B-bentonite | 93.99±0.110 | 3.30 | Jicha et al. (2016) |
| | C-bentonite | 94.022±0.102 | 5.95 | Jones et al. (2021) |
| | D-bentonite | 93.799±0.077 | 6.98 | Jones et al. (2021) |

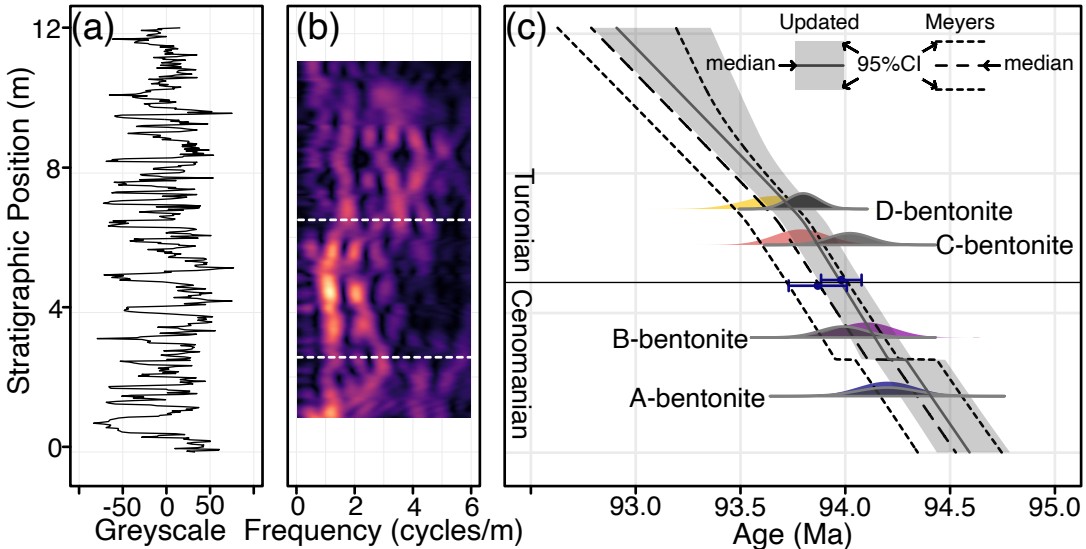

**Figure 6: Results of `astroBayes` modeling of the Bridge Creek Limestone Member greyscale record showing the modeled age of the Cenomanian-Turonian Boundary. A) Bridge Creek Limestone Member grayscale record. B) Evolutive harmonic analysis of panel (2 meter window size, 0.05 meter step size using 3-2π prolate tapers) A) with superimposed layer boundary positions (horizontal dashed white lines). C) Two age-depth models for the Bridge Creek Limestone Member. The colored probability distributions are the dates used for the *Meyers model* and the grey probability distributions are the dates used for the *Updated model*. The blue points and error bars are the `astroBayes` model ages for the Cenomanian Turonian boundary. Note that these points have been slightly offset vertically for visual clarity.**

Results for both the *Meyers* and *Updated* models are shown in Fig. 6C. Evolutive harmonic analysis of the grayscale record after applying the median *Meyers* age-depth model reveals stable eccentricity (~10 cycles/Ma) and obliquity (~20 cycles/Ma) scale frequencies, suggesting that the `astroBayes` age-depth modeling has successfully removed the distortion of these astronomical frequencies due to varying sedimentation rates (Fig. 7). The *Meyers* and *Updated* results are broadly similar and have nearly identical posterior distributions of sedimentation rate (note the parallel model medians in Fig. 6). The *Meyers* model has a wider credible interval compared to the *Updated* model, likely a result of the somewhat more precise radioisotopic dates in the *Updated* model (Table 5). The *Updated* model is also systematically older than the *Meyers* model, showing the influence that the revised bentonite ages have on age-depth model construction. The estimated hiatus durations from both models are similar; the *Meyers* model has a maximum density at 0.023 Myr and the *Updated* model has a maximum density at 0.012 Myr. Both durations are comparable to the duration previously reported in Meyers and Sageman (2004) (0.017 Ma, with uncertainty spanning 0.079 – 0.0254 Ma). Median hiatus durations from `astroBayes` are somewhat longer (*Meyers*- 0.097 Ma; *Updated*- 0.069 Ma) suggesting an eccentricity or precession scale hiatus (Fig. 8). However, the previous estimates of Meyers and Sageman (2004) are explicitly minimum duration estimations and fall within the 95% credible interval of the `astroBayes` modeled duration.

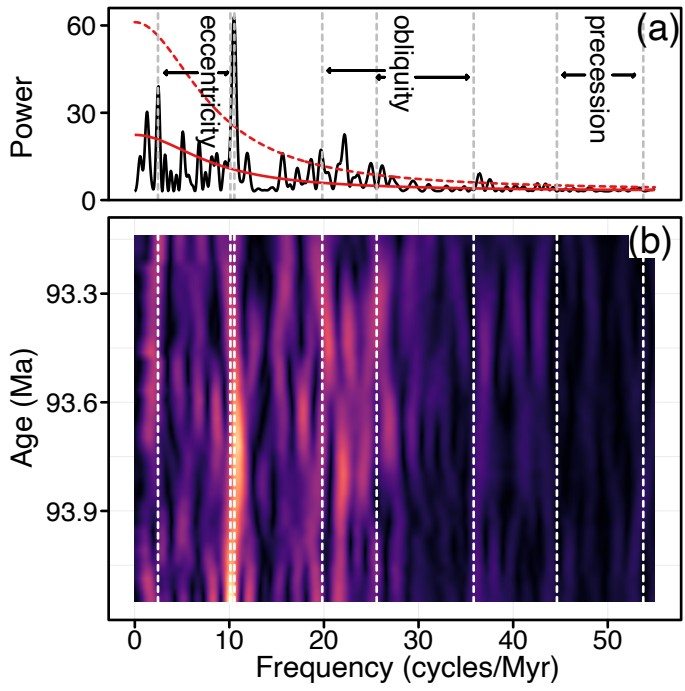

**Figure 7: A)** Periodogram of the Bridge Creek Limestone Member greyscale data after applying the median `astroBayes` age-depth *Meyers model*. The solid red line is the AR1 red noise background and the dashed red line is the standard 95% confidence level (not accounting for multiple-testing). **B)** Evolutive harmonic analysis of Bridge Creek Limestone Member greyscale data after applying the median `astroBayes` age-depth model (0.75 Myr window size, 0.025 Myr step size using 3-2π prolate tapers). In both panels astronomical frequencies (Table 4) used in model construction are shown as vertical dashed lines. Note that in panel B the distortion from variations in sedimentation rate (compared with Fig. 6B) has been removed.

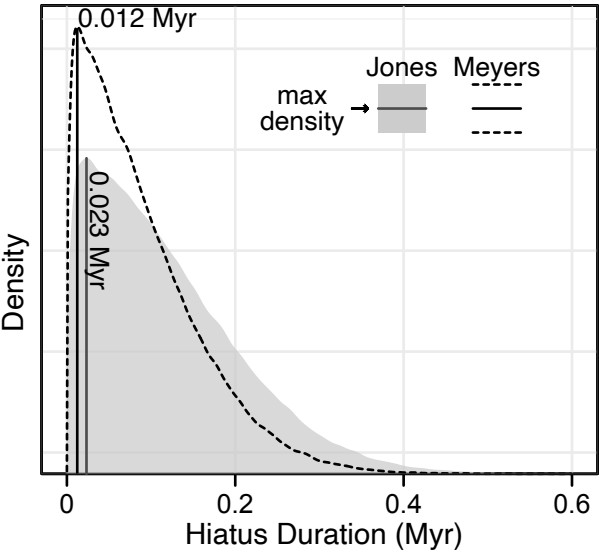

**Figure 8: `astroBayes` modeled duration for the hiatus at the top of limestone marker bed 5 (LS5) in the Bridge Creek Limestone Member.**

Finally, we calculated the age of the Cenomanian-Turonian boundary using both age depth models. The *Meyers* model age for the boundary is 93.87±0.15 Ma (median±95%CI), essentially indistinguishable from the age of 93.90±0.15 Ma reported by Meyers et al. (2012), suggesting that `astroBayes` produces comparable results when using identical data. The *Updated* model boundary-age is slightly older (93.98±0.10 Ma; median±95%CI). The age of the Cenomanian-Turonian boundary has been revised multiple times over the past few years and has variously been reported as 93.9±0.2 Ma (Gale et al., 2020),

93.95±0.05 Ma (Jones et al., 2021), 93.69±0.15 or 94.10±0.15 (Batenburg et al., 2016), and as between 94.007 and 94.616 Ma (Renaut et al., 2023), with most revisions shifting the boundary age older towards about ~94 Ma, a trend that our *Updated* model continues. Both the *Meyers* and *Updated* model-ages are broadly comparable with these previous estimates, although they only slightly overlap with the range of Renaut et al. (2023) (Fig. 9). Crucially however, both `astroBayes` age-depth models provide a continuous record of age for the Bridge Creek Limestone Member that can be used to evaluate geochemical

proxy data and estimate fluxes, interpret the boundary ages and durations of several ammonite biozones present in the section (Meyers et al., 2012, 2001), and foster correlations to other calibrated sections for evaluating mechanisms of Earth System evolution (e.g., Oceanic Anoxic Event 2; (Schlanger and Jenkyns, 1976). Accurate and precise determination of the Cenomanian-Turonian boundary age is important as the boundary serves as an important geochronological marker against which other boundary-ages are determined (Gale et al., 2020).

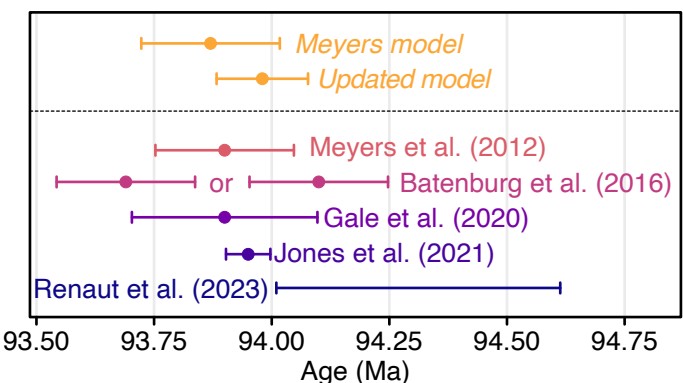

**Figure 9: Modeled `astroBayes` ages and previously reported ages for the Cenomanian-Turonian boundary.**

## 6    Conclusions

Radioisotopic geochronology and astrochronology underly the development of age-depth models that translate stratigraphic position to numerical time. In turn, these models are crucial to the evaluation of climate proxy records and the development of

495 the geologic time scale. Existing Bayesian methods for age-depth modeling generally rely only on radioisotopic dates and as a consequence, do not explicitly incorporate astronomical constraints on the passage of time. However astrochronology is a rich source of chronologic information and its explicit inclusion in the calculation of age-depth models can substantially

improve model accuracy and precision. Here we have presented a new joint Bayesian inversion approach for radioisotopic and astronomical data, `astroBayes`. The method is freely available as an R package and contains a variety of functions for the creation and use of age-depth models including modeling, prediction, and plotting. Our testing shows that `astroBayes` outperforms dates-only age-depth models and produces chronologies that are simultaneously consistent with astrochronology and radioisotopic dates with substantially smaller model uncertainties. Reducing the uncertainty associated with geochronological data, either as discrete dates or age-depth models, allows the testing of cause-and-effect relationships of interrelated climatological and biological events over the course of Earth's history (Burgess and Bowring, 2015; Schmitz and Kuiper, 2013) and has the potential to improve the correlation of geologic events among and between basins worldwide.

## 7    Code and Data Availability

The astroBayes R package and installation instructions are available at github.com/robintrayler/astroBayes. All code and data necessary to reproduce the results of this manuscript (model testing, validation, and case study) are available at github.com/robintrayler/astroBayes_manuscript.

## 8    Author contribution

RBT, and MDS conceived the project and developed the modeling framework with input from SRM. RBT wrote the code for the astroBayes R package and performed testing and validation with input from SRM and MDS. BBS contributed the Bridge Creek Limestone Member grayscale data. RBT, MDS, SRM, and BBS wrote and edited the manuscript.

## 9    Competing Interests

The authors declare that they have no conflict of interest.

## 10    Acknowledgements

We thank Dr. Matthias Sinnesael for providing, and Dr. Christian Zeeden for developing, the Cyclostratigraphy Intercomparison Project CIP2 data used for model testing. We also thank Dr. Jacob Anderson and Dr. Alberto Malinverno for insightful discussions during the development of this project. Finally, we thank Dr. Maarten Blaauw, Dr. David De Vleeschouwer, Niklas Hohmann, and Dr. Matthias Sinnesael for their comments during the open review and discussion of this manuscript. This work was supported by National Science Foundation grants EAR-1813088 (MDS) and EAR-1813278 (SRM).

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

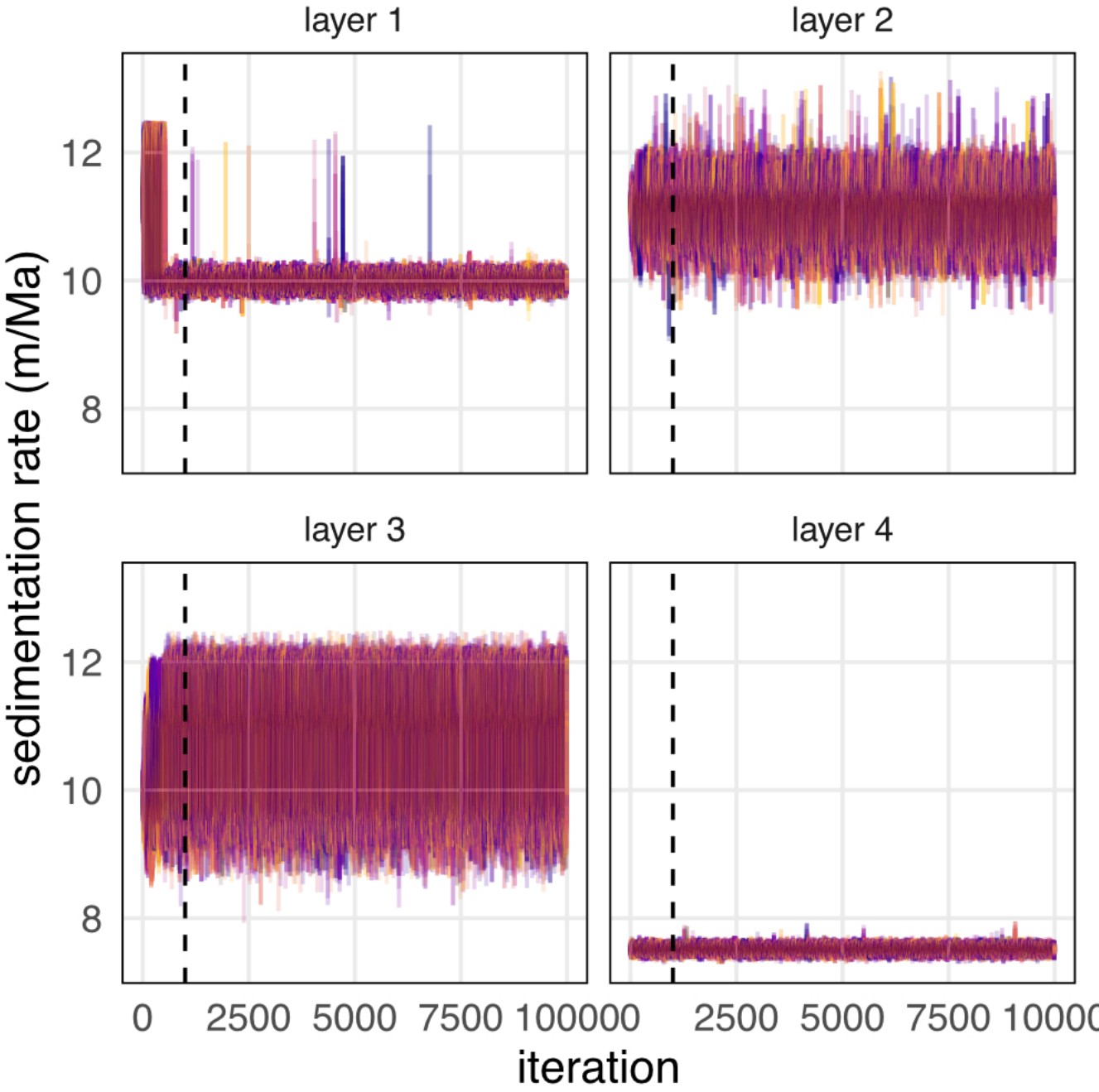

**Figure S1: Superimposed MCMC trace plots of sedimentation rate for 50 randomly chosen models for the TD1 synthetic dataset. Different colors indicate different model runs. The vertical dashed line indicates the burn-in period of 1,000 iterations.**

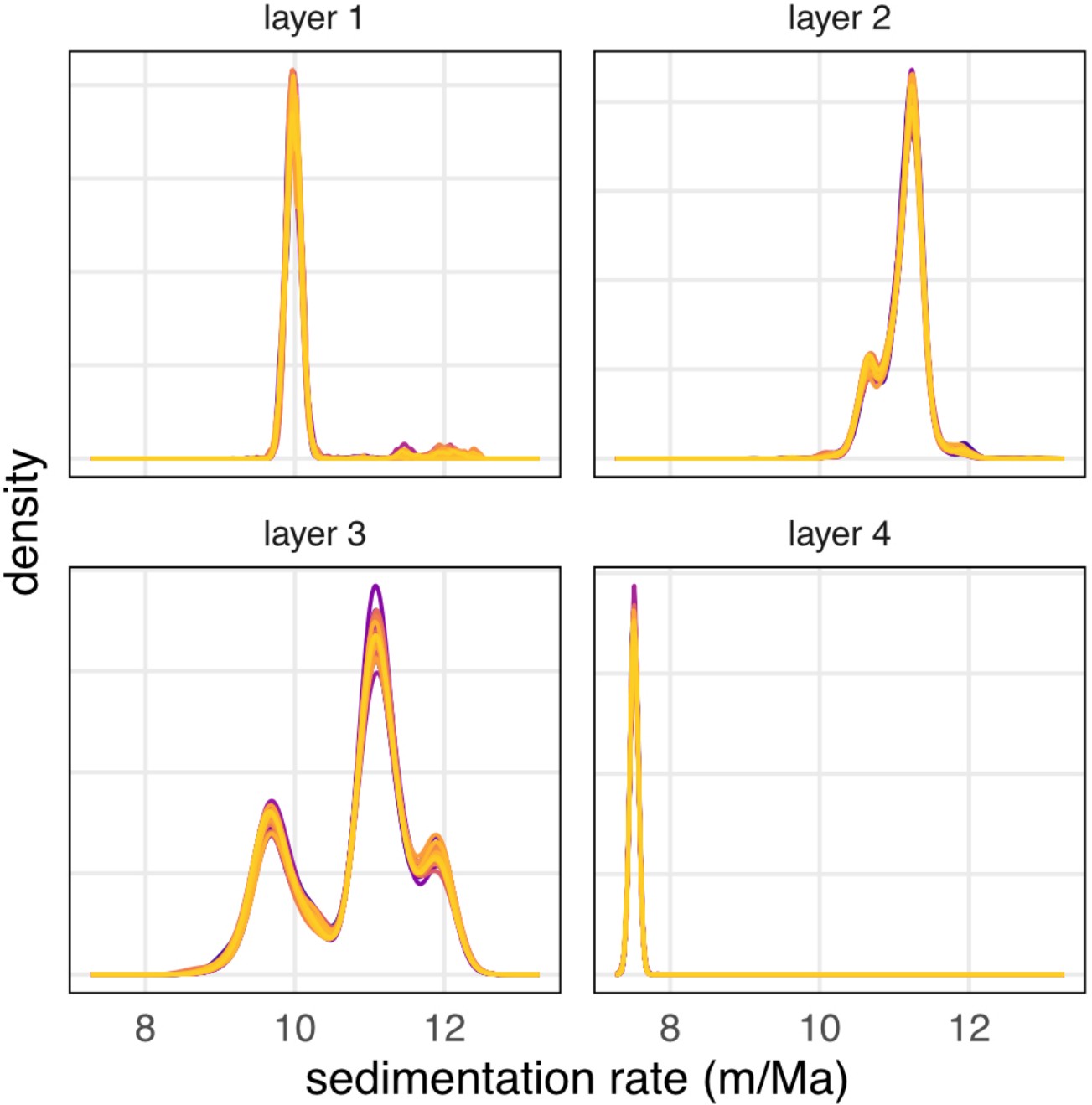

**Figure S2: Superimposed kernel density estimates of the posterior distribution for each model parameter from 50 randomly chosen TD1 validation models. Different colors indicate different model runs.**

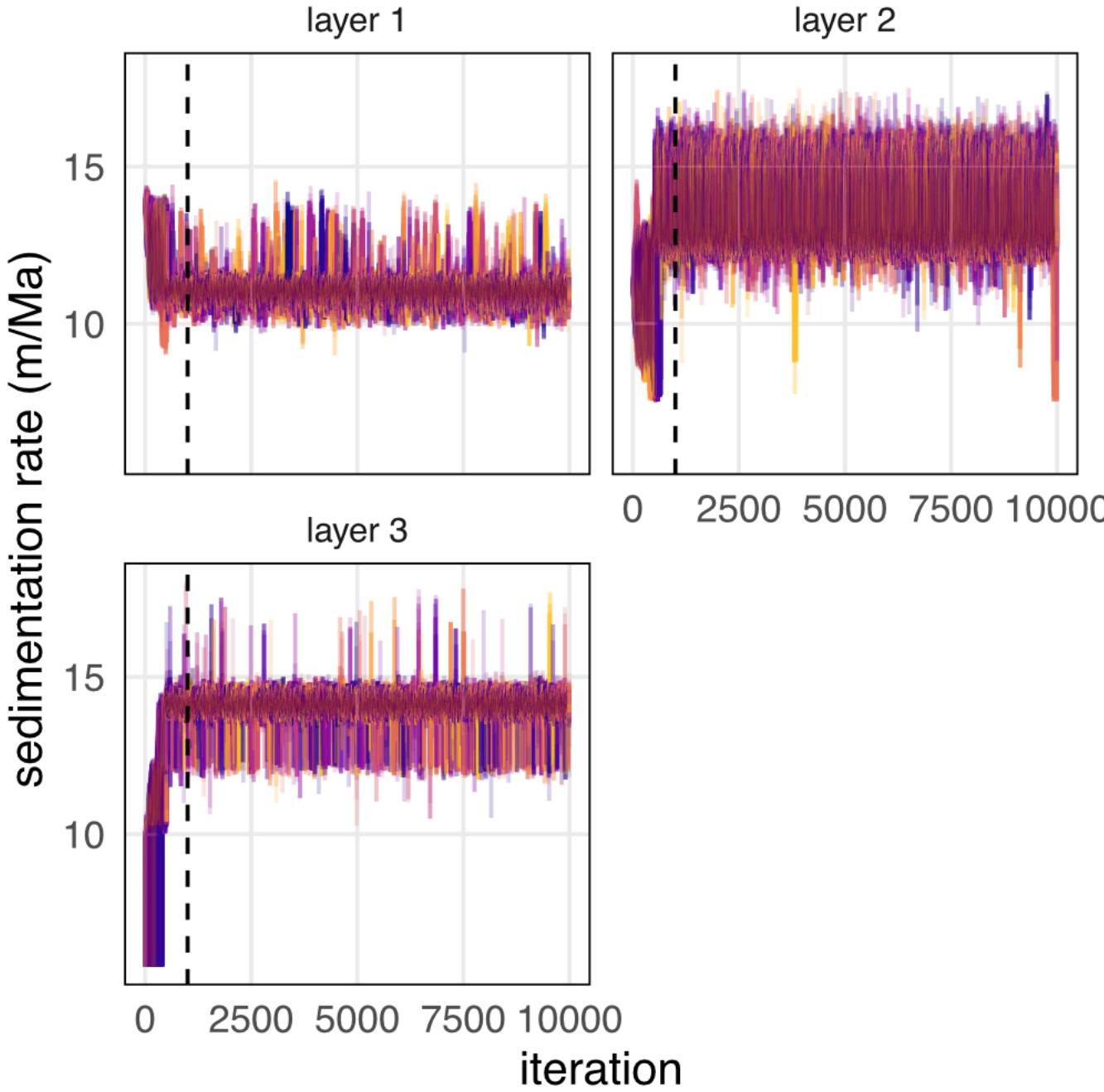

**Figure S3: Superimposed MCMC trace plots of sedimentation rate for 50 randomly chosen models for the CIP2 synthetic dataset. Different colors indicate different model runs. The vertical dashed line indicates the burn-in period of 1,000 iterations.**

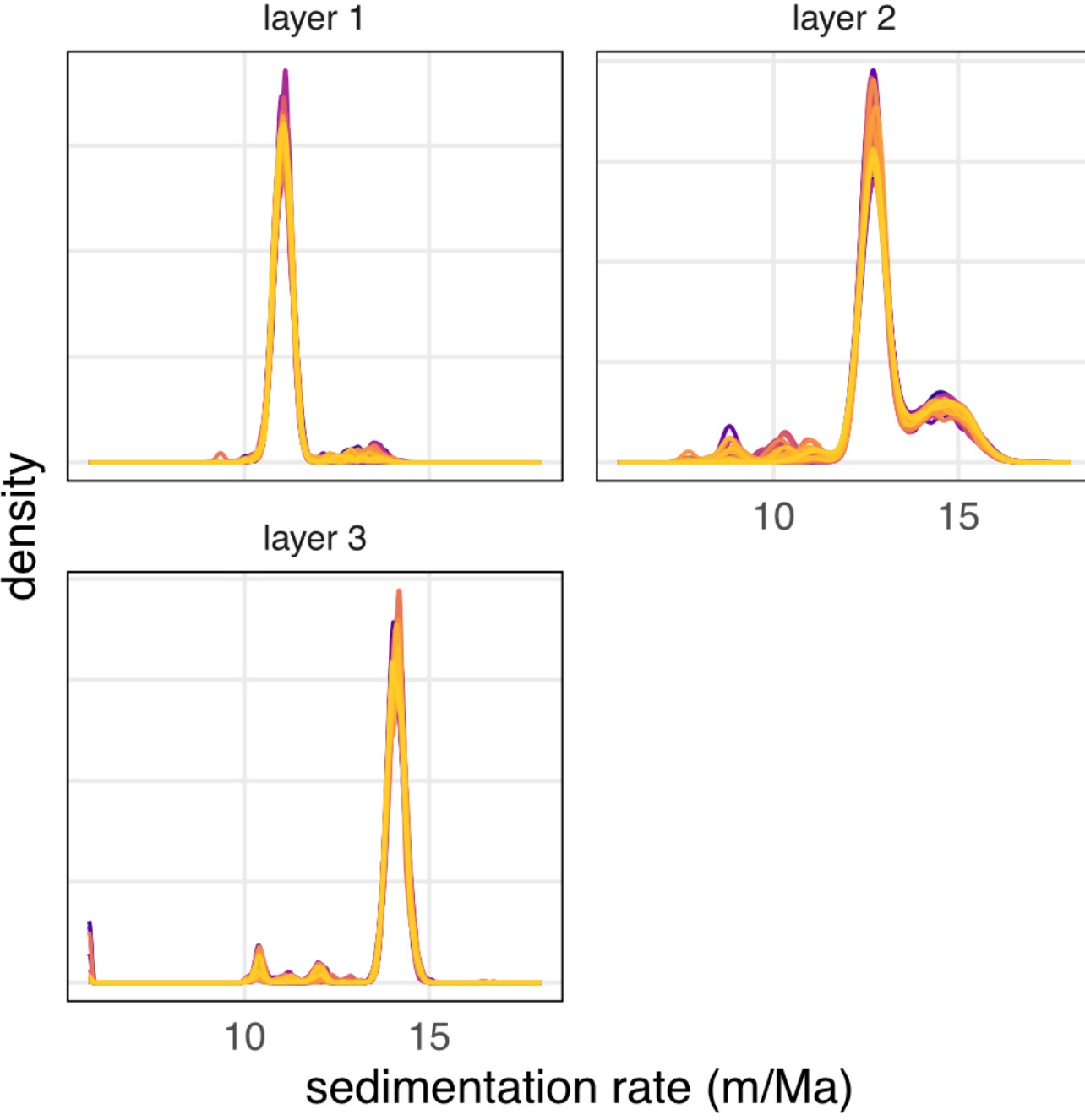

**Figure S4: Superimposed kernel density estimates of the posterior distribution for each model parameter from 50 randomly chosen CIP2 validation models. Different colors indicate different model runs.**