# Peer review of "Bayesian Integration of Astrochronology and Radioisotope Geochronology"

_Geochronology, 2023_

## Referee Comment (RC3)

**Peer review „Bayesian Integration of Astrochronology and Radioisotope Geochronology"**

**David De Vleeschouwer**

In their manuscript, Trayler et al. introduce a novel R package named *astroBayes*, designed for constructing geologic age-depth models that incorporate both radio-isotopic dates and astrochronologic information. To create such a model for a specific section, the user must provide four key pieces of information:

1. A proxy depth-series containing an **assumed** astronomical imprint. At this stage, user input is minimal, and the choice of proxy and its sampling interval is the primary user consideration.
2. Geochronologic dates for the section (stratigraphic position, age, and uncertainty). This input also does not require additional user intervention/decisions.
3. Target frequencies, represented as a vector of astronomical frequencies that are expected to be imprinted in the proxy depth-series mentioned above. The user's input is essential at this stage and likely influences the results in a considerate manner. The potential impact of this user choice becomes evident in the manuscript: The authors made different target frequency choices for the synthetic data sets (Table 2) and the Bridge Creek dataset (Table 4). The different selections raise concerns regarding whether the authors may be favoring certain results by adjusting these frequencies. Notably, the Bridge Creek dataset uses three obliquity periods, despite two of those obliquity components have significantly lower amplitudes compared to the primary 39-kyr obliquity forcing. It also uses only a single precession period, despite precession being influenced by multiple quasi-periodicities.
4. Layer boundaries, representing stratigraphic positions where sedimentation rate changes are expected based on visual inspection of an evolving power spectrum or sedimentological indicators (e.g., hardgrounds, hiatuses, lithology changes). This piece of information is notably user-dependent.

The manuscript is generally well-written and clear. The authors succeed in conveying the general idea behind the algorithm. However, **throughout the manuscript, the authors overlook two critical questions**: First, it remains unclear as to what extent the age-depth model results are **influenced by the user's selection of layer boundaries** (both the number of boundaries and their stratigraphic positions). Second, the authors do not describe the **behavior of the *astroBayes* model when applied to a pure-noise proxy depth-series**.

To investigate the second question, I ran the *astroBayes* model with a purely random noise signal (autoregressive noise with a rho value of 0.9). Apart from the pure-noise character, other depth-series characteristics were similar to the test "cyclostratigraphy" dataset provided in the R package. It appears that, indeed, for a depth-series without an astronomical signal, the age-depth model produces wider uncertainty bands compared to depth-series with an astronomical signal. Nevertheless, these uncertainty bands remain considerably narrower than the "Bchron sausages" referenced in the authors' Figure 3. Obviously, this is because the assumption of piecewise constant sedimentation rates is inherent to the *astroBayes* model. This obviously remains a questionable assumption to make, and to my taste, this assumption does not fully acknowledge true geologic variability in sedimentation rate and the possibility of cryptic hiatuses anywhere in the section. Hence, to my taste, the uncertainty bands for the "pure noise" series in the Figure below seem somewhat over-optimistic, particularly within the interval between bentonite B and C. I recommend that the authors write a dedicated section in the discussion to address this question, explicitly addressing the assumption of piecewise linear interpolation in-between layer

boundaries. This is of paramount importance because the algorithm's user-friendliness can make it highly susceptible to misuse.

[Figure]

 **Figure:** comparison of astroBayes age-depth model result using a signal without (left) and with (right) astronomical imprint.

I was also wondering how the model performs when there is an outlier radio-isotopic date? From what point onward, will astroBayes ignore this outlier? Answering this question will require some sensitivity runs, I assume.

**Minor comments:**

- Line 14: Anchoring chronologies CAN rely on radio-isotopic geochronology… but can also rely on other stratigraphic markers (magnetostratigraphic reversals, biostratigraphic datums, event stratigraphic markers). Are there any ideas about how to incorporate stratigraphic uncertainties on such dates into the astroBayes model?
- Line 28: I find the end of the abstract rather weak. The last sentence does not represent the big "take-home" message for the reader of this paper.
- Line 45: I would recommend a consistent use of Ma and ka for "million years ago" and "thousand years ago" (absolute time, ages). Myr and kyr for "million years" and "thousand years" (durations, relative time differences). In any case, there is no consistent use of these abbreviations throughout the manuscript.
- Line 129 - 148: I would move this part to the end of the Introduction, discussing previous attempts to integrate radio-isotopic dates and astrochronologic interpretations.
- Line 73-77: Repetition of information that was already given in the Introduction.
- Line 82: Wrong Berger et al. citation. You probably mean André Berger et al. 198X or 199X.
- Figure 2f: I can't recognize why the authors drew the horizontal dashed lines (layer boundary positions) at those exact depths. There are no obvious features in the evolutive spectrum that would make me draw them exactly there.
- Line 324 – 325: Not really relevant that future model developments could make the positioning of layers more objective… The required user input in the current version of the algorithm, to me, represents the Achilles heel of your work right now.
- Figure 4: I do not see any points, nor error bars

- Figure 8: Batenburg et al. suggested two tuning options, with an astronomically-tuned age for the C-T boundary of either 93.69 +- 0.15 Ma (Tuning 1) or 94.10 +- 0.15 Ma (Tuning 2).
- Line 396: model → models
- Figure 5: Was the hiatus already known prior to this study? Or was it discovered by astroBayes?
- Figure 6: Which of the two models in Figure 5 are we looking at here? Or is the result in Figure 6 identical for both models in Figure 5?
- Line 466: Case 2 from the Cyclostratigraphic Intercomparison Project was designed by Christian Zeeden, not by Matthias Sinnesael. He should be acknowledged here.

---

## Author Comment (AC1)

**Response to RC1: 'Comment on gchron-2023-22', Maarten Blaauw, 27-Sep-2023**

[1,2,*]Robin B. Trayler    [3]Stephen R. Meyers    [4]Bradley B. Sageman
[2]Mark D. Schmitz

[1]Department of Life and Environmental Sciences, University of California, Merced, CA

[2]Department of Geosciences, Boise State University, Boise ID

[3]Department of Geosciences, University of Wisconsin, Madison, WI

[4]Department of Earth and Planetary Sciences, Northwestern University, Evanston, IL

[*]Corresponding author: rtrayler@ucmerced.edu

- This manuscript proposes to combine proxy data of orbital forcing with radiometric dates in order to produce integrated age-depth models from cores. It builds a piece-wise linear model with constrained accumulation rates and the possibility of hiatuses, and treats the response to the orbital forcing as a constant offset 'a' in years for each section. The method is tested using synthetic and real-world data, and shows huge enhancements in precision compared to dates-only age-models (but see below).

- Generally the method is described well and placed in the wider context through an interesting review of existing methods. However, I would like to see some more detail on how the parameters are estimated, how the priors are set and how these settings affect the age-depth models (robustness analysis).

We thank Dr. Blaauw for their complement about the clarity of our study.

- Section 3.2.3, what limits are put on the accumulation rate, and why are you using a uniform distribution? Why not use a more informative prior on accumulation rate, such as a gamma with a specified mean and shape (the latter can be put at very permissive values, e.g., 1.1)?

We choose to use a uniform distribution as a vague prior distribution so that the cyclostratigraphic likelihoods are the primary control on sedimentation rate. The joint inversion with the radioisotopic dates further limits the possible accumulation rate, and the dates themselves can be used to estimate appropriate bounds for the uniform prior distribution. For example, calculating the slope between each date-pair in sequence gives an average sedimentation rate. Making the same calculation at say ±5σ from the young-tail to the older-tail of a date-pair gives "worst case scenario" sedimentation rates that can be used as bounds for the uniform prior.

- This would then also diminish the likelihood of cycles that require extreme accumulation rates (i.e., the harmonic analysis of Fig. 2c/f would show darker colours for less realistic accumulation rates).

Just to clarify, Panels 2C and 2F are not showing the probability of accumulation rates. Instead they are showing spectral amplitude, calculated over a moving window. These plots are used to interpret stratigraphic "layers" were sedimentation rate is stable but do not inform the absolute value of sedimentation rate necessarily.

- Are there limits on the hiatus size as well, and is the prior also uniform or rather gamma as suggested by Fig. 7?

We did not define a prior distribution for hiatus duration, except for the limitation that hiatus durations be positive values, such that they cannot violate superposition. The probability of a hiatus duration is estimated in the same manner as the other model parameters (anchor age, sedimentation rate(s)).

- The difference in modeled precision between `BChron` and astroBayes is huge, especially in the case where a core has only few radiometric dates. That said, how robust is the assumption of linear accumulation over long time-scales (e.g., the long section of Fig. 3b)? Although this is discussed, I still find it hard to believe that a geological sedimentation process really was exactly linear over large amounts of time - if this assumption is not met, then the reconstructed precision will be illusionary high.

Dr. Blaauw is correct that the increase in precision of `astroBayes` compared to `BChron` results from our choices of a much simpler sedimentation model. We do feel that this choice is justified however, since the stratigraphic information about sedimentation rate that the astrochronology provides, is not trivial and shows that sediment accumulation really can be near-linear for very long periods of time. For example in Figure 5B, it's quite possible to draw a vertical, straight line through the highest amplitude frequency track (~1 cycle/m) within each of the layers we have defined. So while at the very fine scale sedimentation is absolutely a variable process, the astrochronology does show that it can be approximated by a series of linear segments. Clearly this will not always be the case and the suitability of using `astroBayes` to model different datasets will need to be assessed on a case by case basis.

Ultimately our goal is to capture the "true" age model within the `astroBayes` posterior even if we are somewhat simplifying the problem. For example, in figure 2C, the second layer from the base of the section has a varying sedimentation rate that is only partially approximated by our choice of treating it as a single layer. Nevertheless, inspecting the age-depth models in figure 3A-D shows that even when our assumptions of more-or-less constant accumulation are violated the true age-depth model still falls within the 95% credible interval of the posterior, which is reproduced in nearly all cases (see line 300).

- The proposed method uses a limited number of sections of linear accumulation, e.g. <10. Does increasing this to say 50 or 100 shorter linear sections affect the age-depth models by much (e.g., in terms of smoothness and reconstructed uncertainties)? Could you explain a bit more how the 'elbows' (z) between the sections are chosen and how they can be made to vary over depth? This model reminds me of Bpeat (Blaauw and Christen 2005 Applied Statistics 54, 805-816), which modeled accumulation using a handful of linear sections and where the depths of the 'elbows' were part of the parameters to be estimated (this also included hiatuses and outliers).

Currently layer thickness is somewhat constrained by how the probability calculations are made (See line 320 in submitted manuscript).

"… a limitation of our model is that each layer must contain enough time and astrochronologic data to resolve the astronomical frequencies (f) of interest."

Since the layers must contain a a sufficient amount amount of cyclostratigraphic data, increasing the number of model layers degrades the likelihood calculated using equation 2. The cyclostratigraphic sampling rate also plays a role here as the Nyquist frequency within each layer must be high enough to capture each of the target frequencies. Relaxing the first constraint is possible but would require an entirely new modeling framework. Practically this would require a re-write of the core `astroBayes` function and would also greatly increase the computational time required.

It was only briefly mentioned (line 185) in the manuscript, but the implementation in the `astroBayes` package does allow the user to assign uniform uncertainties to the layer boundary positions. The model will randomly adjust the boundary position within the uncertainty for each iteration so that the initial choice of layer position is somewhat less important. We also note that this is the same basic approach that Malinverno et al. (2010) took to deal with constructing Bayesian floating age models from cyclostratigraphic data.

We have expanded part of the *Model Construction* section (lines 176 - 186) so it now reads:

"The selection of layer boundary-positions is an important user defined step, that is informed by detailed investigation of the cyclostratigraphic data. Evolutive harmonic analysis (EHA) is a time-frequency method that can identify changes in accumulation rate by tracking the apparent spatial drift of astronomical frequencies. Expressed as cycles/depth, high amplitude cycles may"drift" towards higher or lower spatial frequencies throughout the stratigraphic record. Assuming these spatial frequencies reflect relatively stable astronomical periodicities, the most likely explanation of those spatial shifts is therefore stratigraphic changes in sedimentation rate (Meyers et al., 2001). That is, stability in spatial frequencies reflects stability in sedimentation rate,

and show that in these cases sedimentation can be approximated by a small number of piecewise linear segments.

We visually inspected EHA plots to develop simple sedimentation models (e.g., 2B) for our testing data sets. We choose layer boundary-positions ($z_1$ – $z_i$) by identifying regions with visually stable spatial frequencies. For example, in 2C, there is a continuous high-amplitude frequency-track between 2-4 cycles/m. Based on visual shifts in this frequency, we choose three layer boundaries, such that this frequency track can be approximated by a vertical line within each layer. In the computation implementation, we also allow the layer layer boundary-positions to vary randomly (within a user specified stratigraphic range) to account for stratigraphic uncertainties in boundary-positions that arise from the fidelity and our inspection of the of the data, similar to the Bayesian cyclostratigraphic approach of Malinverno et al. (2010)."

- What about a potential alternative model, also with set boundaries between the different known sections of nearly but not entirely linear accumulation (e.g., a bit like Bacon but with a very high and strong prior memory on accumulation rate, so a very low variability of accumulation rate over a section, but still some possibility of deviation from an entirely straight line), and with very permissive/wide prior accumulation rates for each of these sections so rates can jump from one z to the next?

This is a **great idea**, that we feel is outside the scope of the current study. Currently, implementing this modeling framework would require rewriting a large chunk of the core `astroBayes` code/functions. However, the model that Dr. Blaauw describes sounds a lot like a hybrid of `bacon` and `astroBayes`, and perhaps future development of the `astroBayes` package could implement an "astroBacon" modeling framework.

- Some more information on the MCMC settings and decisions could be helpful, e.g. in Supplementary Information. Perhaps also provide a short tutorial, much like on the helpful GitHub pages but using the examples of the manuscript and with more information as to what steps are taken and why.

This is a helpful comment, especially from a user-friendliness perspective. We will include the Github tutorial as a vignette in the `astroBayes` R package to make it more discoverable.

- Table 2: The estimates are given as 7 digits, which implies that the length of each of the periods is known to the month (!). Should the values not be rounded to a more realistic precision, and if so, what would that precision be (millennial I'd say)? Are there any estimates of the size and shape of the uncertainties related to the period/frequency estimates of the different orbital cycles?

  Dr. Blaauw is correct that the reported number of decimal places here do not reflect the true precision. There are uncertainties associated with each of the Milankovich frequencies, however, the periods have changed over very long-term time scales (e.g., tens of millions of years), so appropriate periods for say Eocene vs. Pleistocene records will be different. There are various tools available to calculate astronomical periods in deep time (see here) and also (Laskar et al., 2011).

- Does it matter for the harmonic analysis where in time each of the cycles of Table 2 starts?

  No, it does not matter. Since the cyclostratigraphic data is transformed from the stratigraphic position domain into the frequency domain, our method explicitly accounts for the phases of each cycle.

- Section 3.2.2, was no outlier analysis done?

  We thank the reviewer for this suggestion. We did not initially include an outlier analysis, but we have done so now. The sections below have been added to the manuscript in the *Testing and Validation* and *Results* sections respectively. The corresponding R scripts to reproduce the results have likewise been added to the robintrayler/astroBayes_manuscript Github repository.

**Sensitivity To Outlier Ages**

We also tested the sensitivity of `astroBayes` to the inclusion of outlier ages. We repeated the tests from section 3.3.2, with one additional step. After the generation of stratigraphically-randomly

distributed dates, we used Monte Carlo methods to select one date. This date was then randomly adjusted by ±1σ to ±4σ. This creates a date that is either broadly comparable with the underlying true age model (e.g., ±1σ to ±2σ), or outlier ages that may introduce stratigraphic miss-matches (e.g., ±3σ to ±4σ). We choose to introduce these more subtle outliers, since we feel more extreme outlier ages can often be identified and excluded *a priori* based on inspection of the radioisotopic data (Michel et al., 2016). We repeated this procedure 1,000 times using either 2, 4, 6, or 8 dates for each data set (as in the section above), so that 1/2, 1/4, 1/6, and 1/8 dates would be considered an outlier. Each simulation ran for 10,000 MCMC iterations with a 1,000 iteration "burn-in".

**Model Validation**

... `astroBayes` is somewhat sensitive to the inclusion of subtle outlier radioisotopic dates. The inclusion of outlier ages lowered the proportion of the true age-depth model that fell within the 95% credible interval of the `astroBayes` to 89% for TD1, and 88% for CIP2. The relative percentage of outlier ages also does not appear to have a strong influence. ...

**Details**

- Line 13, 49, spatial-temporal? How does is the spatial component involved? Do you rather mean vertical, depth-scale resolution? Spatial could be interpreted as 'horizontal' resolution, as in, how representative is a core of wider spatial events. How would one define high temporal resolution? Rather, mention that this is at, e.g., 10^5-6 yr resolution.

  We have replaced the references to "spatial" with "stratigraphic" which is what we meant.

- 15, high-precision, quantify

  We added "(<±1%)", also see the comment below on the second use of this term on line 94.

- 94, again high-precision - this seems an unnecessary qualifier here as no-one would aim for low precision.

The term "high precision" is often used in deep-time geochronology to distinguish in-situ methods (LA-ICPMS, SIMS) from whole crystal methods. Individual spot analyses from in-situ methods commonly have a precision of ±3-5% and ~±1% precision for weighted means. In contrast, Modern CA-ID-TIMS (U-Pb) and multi-collector mass spectrometers ($^{40}Ar/^{39}Ar$) have a precision of <±1% for single crystal analyses and approach <±0.1% for weighted mean ages. Also see box 1 in (Schmitz and Kuiper, 2013).

- 270, what MCMC thinning was used?

We did not thin our Markov chains. Our understanding is that there continues to be a debate about whether thinning Markov chains is strictly statistically necessary or if mostly used to address computational / computer storage constraints when not thinning would generate unmanageably large vectors or matrices of data, with different studies supporting both conclusions (Link and Eaton, 2012; Owen, 2017). That said, MCMC thinning is a straightforward improvement that can be added to the model code, either simultaneous with model iterations (as in `Bchron::Bchronology()`) or applied post-hoc (as in Dr. Blaauw's own `rbacon::thinner()`), and this can be added to the development version (and integrated into a future version) of the `astroBayes` package.

- 299, The test was done using simulated sections of constant accumulation, so that the model closely follows the simulated truth is no surprise.

We agree that this is not a surprising result but would like to point out that neither of the testing data sets have "constant" accumulation. See the example of the ~10-15 meter layer in panel 5C where the evolutive harmonic analysis shows that sedimentation rate is gradually changing. Furthermore, the sentence this comment refers to is discussing how model results are very consistent, *even* when the number and stratigraphic position of dates is variable which we feel is an important result that would not necessarily be possible with other age-depth modeling frameworks. This is especially apparent in Figure 3 where the sedimentation rate is correctly estimated even in model layers that do not contain any dates, without the inflation in credible intervals seen with `Bchron`.

- 430, to evaluate > Fixed

**References**

Laskar, J., Fienga, A., Gastineau, M., and Manche, H.: La2010: A new orbital solution for the long-term motion of the Earth, Astronomy & Astrophysics, 532, A89, 2011.

Link, W. A. and Eaton, M. J.: On thinning of chains in MCMC, Methods in ecology and evolution, 3, 112–115, 2012.

Malinverno, A., Erba, E., and Herbert, T.: Orbital tuning as an inverse problem: Chronology of the early Aptian oceanic anoxic event 1a (Selli Level) in the Cismon APTICORE, Paleoceanography, 25, 2010.

Meyers, S. R., Sageman, B. B., and Hinnov, L. A.: Integrated Quantitative Stratigraphy of the Cenomanian-Turonian Bridge Creek Limestone Member Using Evolutive Harmonic Analysis and Stratigraphic Modeling, Journal of Sedimentary Research, 71, 628–644, 2001.

Michel, L. A., Schmitz, M. D., Tabor, N. J., Moontañez, I. P., and Davydov, V. I.: Reply to the comment on "Chronostratigraphy and paleoclimatology of the Lodève Basin, France: Evidence for a pan-tropical aridification event across the Carboniferous–Permian boundary" by Michel et al., (2015). Palaeogeography, Palaeoclimatology, Palaeoecology 430, 118–131, Palaeogeography Palaeoclimatology Palaeoecology, 441, 1000–1004, https://doi.org/doi: 10.1016/j.palaeo.2015.10.023, 2016.

Owen, A. B.: Statistically efficient thinning of a Markov chain sampler, Journal of Computational and Graphical Statistics, 26, 738–744, 2017.

Schmitz, M. D. and Kuiper, K. F.: High-Precision Geochronology, Elements, 9, 25–30, 2013.

---

## Author Comment (AC3)

**Response to CC2: 'Comment on gchron-2023-22', Matthias Sinnesael, 17 Oct 2023**

[1,2,*]Robin B. Trayler      [3]Stephen R. Meyers      [4]Bradley B. Sageman

[2]Mark D. Schmitz

[1]Department of Life and Environmental Sciences, University of California, Merced, CA

[2]Department of Geosciences, Boise State University, Boise ID

[3]Department of Geosciences, University of Wisconsin, Madison, WI

[4]Department of Earth and Planetary Sciences, Northwestern University, Evanston, IL

[*]Corresponding author: rtrayler@ucmerced.edu

Dear Trayler et al.,

- Investigating the incorporation of cyclostratigraphic data in Bayesian age-depth models is a very welcomed contribution. Below you can find some minor thoughts I had on what could maybe make some points easier to understand (for me at least):

   We thank Dr. Sinnesael for his comments and complement about the relevance of the study.

1) Around line 155: input is also frequencies, and positions of layer boundaries, could be worth specifying (more clear on GitHub). In general, an example script to run at least one of the cases could be nice for the supplementary information?

   Dr. Sinnesael is correct that the target frequencies are also user-determined. We have expanded section 3.1 slightly to state this.

"…The inputs for `astroBayes` consists of measurements of a cyclostrati­graphic record (*data*) (e.g., $\delta^{18}O$, XRF scans, core resistivity, etc.), and a set of radioisotopic dates (*dates*) that share a common stratigraphic scale. **The user also specifies a set of appropriate target frequencies ($f$; eccentricity, obliquity, precession) for use in probability calculations**…"

2) Somehow indicate the positions of the layer boundary positions on the age-depth plots (e.g. small line on the axis or something)?

   We have added interior tick-marks to the panels in figure 3 that indicate the layer bound­ary positions and updated the figure caption so it now includes:

   "… Interior tick marks on the vertical axis of each panel indicate the layer boundary positions (see also the dashed lines in Figure 2C and 2F)…"

3) Plot the dates from Table 3 on Figure 2?

   We have added the dates to figure 2 as colored PDFs and updated the figure caption so that it now reads:

   "…The colored probability distributions are the synthetic radioisotopic dates used for model stability testing (see Table 3)…"

4) It is nice to see that also the challenge of hiatuses is addressed, but it is important to be very explicit to say that the identification and positioning is user-defined (preferably informed by additional geological context). This is addressed in section 5.2, but I think it would be worth explicitly specifying when you present the CIP2 case that you put the position of the hiatus there because the correct age model is known is this case.

   Dr. Sinnesael makes an  important point to make and is correct that the hiatus position in the CIP2 and Bridge Creek Limestone case study were previously known. We have expanded section 5.2 to include discussion of this point, which now reads:

   "**There are two weaknesses of this approach to estimating hiatus du­ration. First, since hiatus positions are user defined, the stratigraphic**

**position of a hiatus must be known *a priori* and must be informed by geologic (i.e., a visible unconformity) or cyclostratigraphic data (Meyers and Sageman, 2004). In both the CIP2 testing data set and the Bridge Creek Limestone case study (discussed below), the stratigraphic position of the hiatuses were known in advance.** The second weakness is that `astroBayes` cannot reliably estimate durations for hiatuses unconstrained by radioisotopic dates. If a hiatus only has radioisotopic dates stratigraphically above or below, the undated side is unconstrained and duration estimates tend to wander towards an infinite duration. Likewise, if a model layer is bounded by two hiatuses and the layer does not contain any radioisotopic dates, then `astroBayes` cannot reliably resolve the duration of the bounding hiatuses and will tend to"split the difference". However, when hiatuses are well-constrained by radioisotopic dates, `astroBayes` allows the estimation of robust uncertainties of hiatus duration and is a powerful tool when there is external sedimentological or astronomical evidence for hiatuses, as shown in the Bridge Creek Limestone Member case study below."

Best wishes,

Matthias Sinnesael

**References**

Meyers, S. R. and Sageman, B. B.: Detection, quantification, and significance of hiatuses in pelagic and hemipelagic strata, Earth and Planetary Science Letters, 224, 55–72, 2004.

---

## Author Comment (AC4)

**Response to CC1: 'Comment on gchron-2023-22', Niklas Hohmann, 11 Oct 2023**

[1,2,*]Robin B. Trayler        [3]Stephen R. Meyers        [4]Bradley B. Sageman

[2]Mark D. Schmitz

[1]Department of Life and Environmental Sciences, University of California, Merced, CA

[2]Department of Geosciences, Boise State University, Boise ID

[3]Department of Geosciences, University of Wisconsin, Madison, WI

[4]Department of Earth and Planetary Sciences, Northwestern University, Evanston, IL

[*]Corresponding author: rtrayler@ucmerced.edu

- The manuscript presents a Bayesian approach to estimate age-depth models from cyclostratigraphic and radiometric information. The method is implemented in an R package, and applied to synthetic and empirical examples. A highlight of the method is to incorporate information on hiatuses.

- Code availability and documentation are excellent, and meet best practices in research software development.

  Thank you.

- I have some comments regarding the package and how it could be improved (see below). Given the already very high level of code quality these comments are minor.

- The authors use a Bayesian approach to estimate age-depth models. This mathematical part would profit from more technical details to document the model and the inner workings of the package. E.g., merging Eqs. 2 & 3 would make the model more explicit and easier to connect

it with the provided code. A justification of the choice of priors and the MCMC algorithm as well as a discussion of computation time and convergence of the MCMC method should be added to the text (or in the supplementary material).

Dr. Blaauw made a similar comment in Referee Comment 1, and we have added some discussion there on the justification of the choice of a uniform prior distribution as well as some guidance on choosing appropriate prior values.

As far as the performance of the MCMC algorithm, computational time and convergence can vary quite a bit in our experience depending on many factors, including the number of model layers, the strength (e.g., signal:noise) of the astronomical data, and the precision of the radioisotopic dates. As such it is difficult for us to give any general advice on the appropriate length of MCMC chains since different problems will require different settings as choosing MCMC chain lengths is somewhat of a heuristic process.

**Double use of cyclostratigraphic information**

• Cyclostratigraphic information is used twice in the analysis: Once before the Bayesian analysis to visually identify the breakpoints in sedimentation rate, and then in the Bayesian analysis to estimate the sedimentation rate & age depth model. Intuitively it is not obvious how this re-use of data influences the outputs. If the break point are determined based on spatially stable frequencies, how informative can they still be for the Bayesian analysis? E.g. in the extreme case where the visual inspection shows no change in sedimentation rates between two tie points, it is not straightforward to see how much information the approach adds. A brief discussion of the relation between the two steps of analysis and how one influences the other (or why they are independent) would help clarify this.

The reviewer is correct that the cyclostratigraphic data is used twice, once to estimate breakpoint positions (layer boundaries) and later as part of the MCMC estimation of sedimentation rate. We do not belie this is "reuse" of the data though. Determining the layer boundary positions by identifying stratigraphically stable frequencies only identifies that the sedimentation rate is stable within that layer. The second step, the full

MCMC model, estimates the rate within the layer. If "the visual inspection shows no change in sedimentation rates between two tie points" then in our view, that would justify the placement of the two layer boundaries since they are defining a zone of stable sedimentation.

**Comparison with BChron and assumptions on sedimentation rates**

- The authors show figures with uncertainties from their approach and those derived from BChron, and briefly discuss the different assumptions made by both methods. They conclude that "astrochronology provides a clear, strong constraint on the stratigraphic variability in sedimentation rate" and "astrochronology […] can substantially improve [age-depth] model accuracy and precision"

- This is potentially misleading, as BChron has very loose assumptions on variability in sedimentation rates, while astroBayes has piecewise constant sedimentation rates "baked in". Naturally, this assumption limits the uncertainty the model can display between the radiometric dates. An example demonstrating that the reduced uncertainty is generated by the information added by cyclostratigraphic data, and not by the model assumption of piecewise constant sedimentation rates, would greatly strengthen the authors point.

This is similar to a critique made by Dr. De Vleeschouwer in Referee Comment 2. We agree that the improvement in uncertainties *is because* our choice of a simpler sedimentation model naturally limits variability relative to `Bchron`. We feel that the second point here - demonstrating that the reduction comes from the astrochronology - is a little difficult to test, however. The astrochronology is what allows us to use a simple sedimentation model, and removing the information added by it would necessarily mean that we should not use such a model. in other words, in the absence of informative astrochronologic data, `astroBayes` would not be an appropriate tool. Nevertheless, expanding the discussion of this is very relevant to the manuscript and we will do so.

**Comments on the astroBayes package & repository**

**Software citation:**

- The package itself uses other scientific software (e.g. `astrochron`). This should be made explicit in the main manuscript by stating the dependencies and citing the used packages.

  `astroBayes` relies on several established R packages including `astrochron` to calculate periodograms and manipulate the astronomical data. It also relies on various `tidyverse` packages for data manipulation and plotting. The package dependencies are documented in the package `DESCRIPTION` file. Since most of these packages are used "under-the-hood", we feel that they do not need to be explicitly mentioned in the main manuscript, but can be documented as is and in the GitHub `README` and the (to be added) package vignette.

- Based on the README, the package is deposited on Zenodo and assigned a DOI. This is excellent, and should be mentioned in the main text. The package should also be cited in the main text to specify on which version the analyses were run, and increase computational reproducibility.

  We will add appropriate citations to the Zenodo DOI's immediately before revision. Since these comments and others have necessitated making some changes to both the `astroBayes` and `astroBayes_manuscript` repositories both will get new releases/DOI's once the revision process is finished. The `astroBayes` GitHub repository will remain under active development as we add more capabilities in the future.

- Citation info generated using `citation("astroBayes")` does not show the DOI. I suggest adding it in there to have a tighter association between the package and the archived version.

  We have added a citation to the draft version of the manuscript to the `astroBayes` package (commit 10b517bd7711d3cf9cb35e7e6368dde4a619790e). Currently it cites the preprint version and DOI but we will update this after the review process is complete.

- I suggest to add a CITATION.cff file to the repository (https://citation-file-format.github.io/) so the package can be directly cited from GH.

We have added a citation file to the `astroBayes` repository. (commit 10b517bd7711d3cf9cb35e7e6368dde4a

We will update this as necessary after the publication process is complete.

**Examples**

- The example provided in the README runs smoothly and is instructive. From a packaging perspective I recommend moving it to a vignette so it is directly associated with the package and also available to non-GH users.

  This is similar to a comment made by Dr. Blaauw in RC1. We will add a vignette to the R package with a fully worked example.

- Package installation from GH works, but `devtools::check()` throws an error due to missing package dependencies. Fixing this is a requirement for submitting the package to CRAN (which I highly recommend)

  Thank you for catching this. We have fixed the missing package dependencies (see commit `bd600b4209a2ee828f5efd726ed348be0dc0379c` to the `astroBaye` GitHub repository). CRAN submission is planned for some time after paper acceptance.

- `summary(age_model, type = 'hiatus')` does not return anything. I think it'd be helpful if it returned that there is no hiatus in the age-depth model (which is relevant information)

  This was a bug and has been fixed. see commit `bd600b4209a2ee828f5efd726ed348be0dc0379c` to the `astroBayes` GitHub repository.

**Comments on the `astroBayes_manuscript` repository**

- Running instructions are present, but requires to execute scripts in a specific order. Will the outputs still be the same if the scripts are executed in a different order, or will the scripts break? If so, it might be worth having a higher-level script that ensures everything is executed in the right order.

  The first 4 scripts (`_stability.R`, `_validation.R`) can be run in any order but must be run before the remaining scripts. We can add an additional script that uses `source()`' to run all the scripts in the appropriate order.

- I am torn regarding the computational reproducibility of the study. As the data generated by the code is not available, I cannot reproduce the figures. Based on the estimated run time of a week it is also not feasible to produce the data on my machine. This is not a problem with the study itself, but rather Bayesian approaches in general: Computation time is too long to generate data from scratch, and the amount of data generated is too large to be easily archived. I am unsure how or if this can be resolved, but the runtime and amount of data generated should be mentioned in the manuscript.

  We agree that making the full results and testing data available is a challenge since they total ~1.5 Tb of data. Currently there are 10,000 age-depth model outputs in the testing data set. This is a lot of data, but it is not completely out of the question to generate these models on a personal computer. All model testing was done on a 2023 Mac Studio with 64Gb of RAM, a 2 Tb hard drive, and an Apple M1 Ultra processor. This is a powerful computer but most academic researchers likely have access to a computer with similar specs. We do note that it is very feasible for an individual to use the code in the `astroBayes_mansuscript` repository to generate a smaller number of models on a personal computer. In most cases the scripts have a single line of code that sets the number of models to generate a smaller (but still useful) number. This in itself is a good test. Since we are using a Bayesian approach, independently generated models should reproduce our results.

  That said, we are currently looking into solutions to host the original testing outputs for public download. Again this doesn't solve the size problems (~1.5 Tb) but it would make the exact testing data available.

- As the discussion on the manuscript continues, it might be worth to make releases of the manuscript repo to make it clearer to which version of the manuscript the comments refer to.

  This is the plan. We will revise and commit changed to the manuscript that correspond to each reviewer/ commenter.

---

## Author Comment (AC5)

**Response to RC1: comment on gchron-2023-22', David De Vleeschouwer, 13 Oct 2023**

[1,2,*]Robin B. Trayler        [3]Stephen R. Meyers        [4]Bradley B. Sageman

[2]Mark D. Schmitz

[1]Department of Life and Environmental Sciences, University of California, Merced, CA

[2]Department of Geosciences, Boise State University, Boise ID

[3]Department of Geosciences, University of Wisconsin, Madison, WI

[4]Department of Earth and Planetary Sciences, Northwestern University, Evanston, IL

[*]Corresponding author: rtrayler@ucmerced.edu

- In their manuscript, Trayler et al. introduce a novel R package named `astroBayes`, designed for constructing geologic age-depth models that incorporate both radio-isotopic dates and astrochronologic information. To create such a model for a specific section, the user must provide four key pieces of information:

- A proxy depth-series containing an **assumed** astronomical imprint. At this stage, user input is minimal, and the choice of proxy and its sampling interval is the primary user consideration.

Dr. De Vleeschouwer has highlighted an important point that we should have made explicit in the manuscript. That is, `astroBayes` is *not an astrochronologic testing method.* Statistical testing for the presence of an astronomical signal must be done using other hypothesis-testing approaches (e.g., see Meyers (2019); Sinnesael et al. (2019)) before age-depth modeling with `astroBayes`. `astroBayes` is most similar to the frequency domain Bayesian approach of Malinverno et al. (2010), which does not conduct statistical testing (e.g., no *p*-value is calculated; see also the time-domain tuning approach

of Lisiecki and Raymo (2005)). In our view, `astroBayes` is intended to be the end-point of an astrochronologic workflow not the beginning. Text will be added to highlight these points.

- Geochronologic dates for the section (stratigraphic position, age, and uncertainty). This input also does not require additional user intervention/decisions.

  Dr. De Vleeschouwer is correct that this step does not require additional user intervention, but we will highlight that we are assuming that the user will use "good" dates that have already been screened for outliers or anomalies, which may arise from geologic processes such as open system behavior (e.g., loss of daughter product).

- Target frequencies, represented as a vector of astronomical frequencies that are expected to be imprinted in the proxy depth-series mentioned above. The user's input is essential at this stage and likely influences the results in a considerate manner. The potential impact of this user choice becomes evident in the manuscript: The authors made different target frequency choices for the synthetic data sets (Table 2) and the Bridge Creek dataset (Table 4). The different selections raise concerns regarding whether the authors may be favoring certain results by adjusting these frequencies. Notably, the Bridge Creek dataset uses three obliquity periods, despite two of those obliquity components have significantly lower amplitudes compared to the primary 39-kyr obliquity forcing. It also uses only a single precession period, despite precession being influenced by multiple quasi-periodicities.

  Dr. De Vleeschouwer is correct that the choice of target frequencies can potentially have a substantial influence on the `astroBayes` posterior. We choose different target frequencies for the testing data and case study for two reasons. First, both testing data series (TD1 and CIP2) were designed to mimic late-Quaternary records, while the Bridge Creek Limestone section is Late Cretaceous. The target frequencies used for with the TD1 and CIP2 testing data sets were calculated using the Laskar et al. (2004) solution for precession and obliquity from 0-10 Ma, and the Laskar et al. (2011) LA10d solution for eccentricity from 0-20 Ma.The Late Cretaceous precession and obliquity terms were calculated using the reconstruction of Waltham (2015). The target frequencies used for the Bridge Creek Limestone case study were chosen from two sources. The precession and obliquity terms were calculated from the reconstruction of Waltham (2015). The eccentricity terms were based on the LA10d solution (Laskar et al., 2011) from 0-20 Ma (the short and long eccentricity periods are not expected to undergo persistent long term drift, as is the case for precession and obliquity). We included the additional ~0.050 Myr and ~0.028 Myr obliquity periods (based on the Waltham (2015) "k" estimate) for the Cretaceous, since these periods have previously been reported for this section (Sageman et al., 1997; Meyers et al., 2001). The choice to use a singe precession term was based on an observation that multiple precession terms lead to a multimodal likelihood function that disagreed with previous sedimentation rate estimates for the Bridge Creek Limestone (Meyers et al., 2001). However we recognize that this was a qualitative decision on our part and we will investigate this further during revision. We will re-run these analyses using different combinations of precession terms (e.g., averaging or including both ~0.018 Ma terms), to test if they significantly influence modeling results

We agree that it is puzzling that such a strong ~0.050 Myr cycle is observed in the data, although this appears to be a feature of other contemporaneous records, such as at DSDP Site 603B, Tarfaya S13, and ODP Site 1261B (Kuhnt et al., 1997; Meyers et al., 2012b). It is possible, for example, that an Earth-System process is amplifying the ~0.050 obliquity response, and/or that it is impacted by oceanographic processes such as outlined in Wallmann et al. (2019).

- Layer boundaries, representing stratigraphic positions where sedimentation rate changes are expected based on visual inspection of an evolving power spectrum or sedimentological indicators (e.g., hardgrounds, hiatuses, lithology changes). This piece of information is notably user-dependent.

- The manuscript is generally well-written and clear. The authors succeed in conveying the general idea behind the algorithm. However, throughout the manuscript, the authors overlook two critical questions: First, it remains unclear as to what extent the age-depth model results are influenced by the user's selection of layer boundaries (both the number of boundaries

and their stratigraphic positions). Second, the authors do not describe the behavior of the astroBayes model when applied to a pure-noise proxy depth-series.

- To investigate the second question, I ran the astroBayes model with a purely random noise signal (autoregressive noise with a rho value of 0.9). Apart from the pure-noise character, other depth-series characteristics were similar to the test "cyclostratigraphy" dataset provided in the R package. It appears that, indeed, for a depth-series without an astronomical signal, the age-depth model produces wider uncertainty bands compared to depth-series with an astronomical signal. Nevertheless, these uncertainty bands remain considerably narrower than the "Bchron sausages" referenced in the authors' Figure 3. Obviously, this is because the assumption of piecewise constant sedimentation rates is inherent to the astroBayes model. This obviously remains a questionable assumption to make, and to my taste, this assumption does not fully acknowledge true geologic variability in sedimentation rate and the possibility of cryptic hiatuses anywhere in the section.

We appreciate this comment. We have more comments on the random-noise test below, but would like to point out that Dr. De Vleeschouwer's first point about the piecewise linear model is explicitly discussed in section 5.1 of the manuscript, starting around line 525. We feel that this discussion, paired with the added discussion below should provide enough guidance for users to know when a piecewise linear accumulation model is appropriate (or not). We also note that simpler sedimentation models have often been used in the past to approximate accumulation (Malinverno et al., 2010; Meyers, 2019).

- Hence, to my taste, the uncertainty bands for the "pure noise" series in the Figure below seem somewhat over-optimistic, particularly within the interval between bentonite B and C. I recommend that the authors write a dedicated section in the discussion to address this question, explicitly addressing the assumption of piecewise linear interpolation in-between layer boundaries. This is of paramount importance because the algorithm's user-friendliness can make it highly susceptible to misuse.

We appreciate the thought that Dr. De Vleeschouwer has put into this review and are especially glad that he has provided an example noise series analysis. He raises an important point that we did not really make clear in the manuscript. `astroBayes` is intended to be used after the cyclostratigraphic data has been vetted and shown to contain statistically significant astronomical signals through other means (e.g., null-hypothesis testing). We were able to skip this step for the synthetic testing data since they were generated directly from astronomical solutions, and for the Cenomanian-Turonian case study since this section has been repeatedly investigated over the past c. 20 years, including with the Average Spectral Misfit astrochronologic testing approach (see Fig. 7 & of Meyers and Sageman (2007)) (Sageman et al., 1997, 1998; Meyers et al., 2001, 2012a; Meyers and Sageman, 2004).

To address Dr. De Vleeschouwer's comments, we have added a section cautioning about the appropriate use and potential misuse of `astroBayes`. We now include a similar noise series example as that provided by Dr. De Vleeschouwer (see Figure 1 below) and have provided some guidance on when astroBayes is an inappropriate tool. New text underscores that the use of time-frequency analysis to assess bedding stability in specific layers is requisite for evaluation of the underlying simplifying assumption of piecewise-linear sedimentation rates. We also note that the astroBayes approach is robust to moderate departures from this assumption, as noted in the response to Dr. Blaauw's review:

> "Ultimately our goal is to capture the"true" age model within the `astroBayes` posterior even if we are somewhat simplifying the problem. For example, in figure 2C, the second layer from the base of the section has a varying sedimentation rate that is only partially approximated by our choice of treating it as a single layer. Nevertheless, inspecting the age-depth models in figure 3A-D shows that even when our assumptions of more-or-less constant accumulation are violated the true age-depth model still falls within the 95% credible interval of the posterior, which is reproduced in nearly all cases (see line 300)."

[Figure]

Figure 1: Results of `astroBayes` modeling of the TD1 testing dataset, with the cyclostratigraphic data replaced by randomly generated red-noise. A) Randomly generated red-noise B) Age-depth model generated using the correct dates, frequencies, and layer boundaries, and the red-noise cyclostratigraphic data C) Evolutive harmonic analysis of A). The dashed lines indicate the layer boundary positions used for other model testing (see Figure 2 in the manuscript). The arrows indicate the uncertainty in layer boundary position since the data lacks any stratigraphically stable and continuous frequencies.

**Misuse of `astroBayes`**

"Because `astroBayes` is available as an R package, it is straightforward to install and use, assuming familiarity with the R programming language (R Core Team, 2023). Given this, we feel we should discuss appropriate and inappropriate use of the modeling framework. First, `astroBayes` is not a method to test for the presence of statistically-significant astronomical signals and it does not include any null-hypothesis tests. There are a variety of statistical methods available to test for the presence of astronomical signals in the rock record (Huybers and Wunsch, 2005; Meyers and Sageman, 2007; Zeeden et al., 2015; Meyers, 2019) which should be used prior to `astroBayes` modeling. Instead, `astroBayes` is intended to be used to develop age-depth models after the presence of astronomical signals has been established using other methods. Similarly, `astroBayes` does not include automated outlier rejection for radioisotopic dates (Bronk Ramsey, 2009) and these data should be pre-screened following best practices for high precision geochronology (Michel et al., 2016; Schmitz and Kuiper, 2013).

    `astroBayes` is software, and it is quite possible to generate an age-depth model from data

that lacks any astronomical signals or contains outlier radioisotopic dates. Therefore `astroBayes` makes three assumptions about the input data. First, the cyclostratigraphic *data* has been vetted and has been shown to contain statistically significant astronomical signals using other astrochronologic testing approaches. 2) The user-specified layer boundary positions ($z$) have been informed by either careful inspection of the cyclostratigraphic *data* (e.g., time-frequency analysis such as EHA), and other geologic data (e.g., visible facies changes), or both. 3) The radioisotopic dates have been prescreened and do not contain obvious outlier dates or violations of fundamental geologic principles (e.g., superposition).

For a simple example of an inappropriate use of `astroBayes`, we replaced the cyclostratigraphic *data* in the TD1 data set with randomly generated red-noise. All other parameters (dates, layer boundaries, target frequencies) remained the same (see: Figure 2, Table 2 and Table 3 in manuscript). Together, we used these data to generate an `astroBayes` age-depth model, shown in Fig. 1. The resulting age-depth model (Fig. 1 B) looks superficially similar to the example models shown in Fig. **??**. Since the radioisotopic dates still offer some limits on sedimentation rate, the median model still appears similar to the true age model. While the model credible interval is somewhat wider, notably, it does not "balloon" and the overall uncertainties remain low compared to dates-only models (e.g., `BChron`. However, while this age-depth model looks superficially promising, it violates two of the assumptions discussed above. First, the "cyclostratigraphic" *data* (red-noise) does not contain any statistically significant astronomical periods, leading to meaningless probability calculations. Second, because the "cyclostratigraphic" *data* is random, it cannot be used to inform the placement of layer boundaries. Indeed the evolutive harmonic analysis shown in Fig. 1 C shows no stratigraphically stable frequencies, making the layer boundary positions used for this example arbitrary and incorrect. The `astroBayes` modeling framework explicitly assumes a piecewise linear sedimentation model (Figure 1 in manuscript) where sedimentation rate only varies at layer boundaries but is otherwise stable. Since for this example the "cyclostratigrapy" contains no astronomical signals, and the layer boundary positions cannot be reliably determined, `astroBayes` would be an inappropriate modeling tool."

- I was also wondering how the model performs when there is an outlier radio-isotopic date? From what point onward, will astroBayes ignore this outlier? Answering this question will

require some sensitivity runs, I assume.

This concern was also raised by Dr. Blaauw in Referee Comment 1 and has been addressed there. Briefly, we have added a sensitivity analysis that includes outlier dates. The model is somewhat sensitive to the inclusion of outliers but the proportion of the true-age model contained by the `astroBayes` credible interval is still ~87-90%.

**Minor comments**

- Line 14: Anchoring chronologies CAN rely on radio-isotopic geochronology… but can also rely on other stratigraphic markers (magnetostratigraphic reversals, biostratigraphic datums, event stratigraphic markers). Are there any ideas about how to incorporate stratigraphic uncertainties on such dates into the astroBayes model?

  Although it's not explicitly stated in the paper, there is an option for radioisotopic dates to be assigned a stratigraphic "thickness" which is treated as a uniform stratigraphic uncertainty in astroBayes. The modeling algorithm randomly adjusts the stratigraphic position (with in the bounds) of the date each iteration to account for this. This does allow for "stratigraphic uncertainty". As for the other anchor types, if they can be expressed as an age±uncertainty that is Gaussian, then they can be included in the same way as the radioisotopic dates. Other distribution types (gamma, uniform, etc.) would require some modification of the modeling code.

- Line 28: I find the end of the abstract rather weak. The last sentence does not represent the big "take-home" message for the reader of this paper.

  We have edited this to read:

  "Finally, we present a case study of the Bridge Creek Limestone Member of the Greenhorn Formation where we refine the age of the Cenomanian-Turonian Boundary, showing the strength of this approach when applied to deep-time chronostratigraphic questions."

- Line 45: I would recommend a consistent use of Ma and ka for "million years ago" and "thousand years ago" (absolute time, ages). Myr and kyr for "million years" and "thousand years" (durations, relative time differences). In any case, there is no consistent use of these abbreviations throughout the manuscript.

  We have changed all references to Ma / Myr as appropriate. All durations now use Myr (long eccentricity = 0.405 Myr) while numerical ages remain as Ma (C-T boundary age = 94 Ma).

- Line 129 - 148: I would move this part to the end of the Introduction, discussing previous attempts to integrate radio-isotopic dates and astrochronologic interpretations.

  We would prefer to keep this section where it so manuscript introduction remains short, without getting into the weeds.

- Line 73-77: Repetition of information that was already given in the Introduction.

- Line 82: Wrong Berger et al. citation. You probably mean André Berger et al. 198X or 199X.

  Fixed. This was intended to cite Berger et al. (1992).

- Figure 2f: I can't recognize why the authors drew the horizontal dashed lines (layer boundary positions) at those exact depths. There are no obvious features in the evolutive spectrum that would make me draw them exactly there.

  The lower layer boundary was placed to correspond to the position of the known hiatus in the CIP 2 dataset. The upper layer boundary was chosen to correspond to the bifurcation of the 1-2 cycles/m frequently track and the "smearing" of the 3-4 cycles/m track.

- Line 324 – 325: Not really relevant that future model developments could make the positioning of layers more objective… The required user input in the current version of the algorithm, to me, represents the Achilles heel of your work right now.

It is our intention to continue development of the astroBayes package but Dr. De Vleeschouwer is correct that future revisions are not relevant to this present manuscript. However, we do not agree that the user-specification of layer boundaries is an "Achilles heel" as it informed based on cyclostratigraphic evaluation (e.g., EHA analysis) and other geologic data (e.g., facies changes), and can be addressed in sensitivity tests with iterative `astroBayes` analyses. We do recognize that this layer boundary specification makes developing an astroBayes age-depth model more involved than using `rbacon` or `BChron`. However, when used appropriately we feel that `astroBayes` is a powerful tool with capabilities not found in other approaches.

- Figure 4: I do not see any points, nor error bars

  Could this be a PDF rendering error? There are a lot of points / error bars on the figure. They do overlap and blur together but they are clearly visible to us.

- Figure 8: Batenburg et al. suggested two tuning options, with an astronomically-tuned age for the C-T boundary of either 93.69 +- 0.15 Ma (Tuning 1) or 94.10 +- 0.15 Ma (Tuning 2).

  We have added these ages to Figure 8 and to the manuscript text.

- Line 396: model à models

  Fixed

- Figure 5: Was the hiatus already known prior to this study? Or was it discovered by as-troBayes?

  This hiatus was originally identified by Meyers and Sageman (2004) (see lines 375 and 398 of the manuscript) who also provided an estimation of its duration. We allowed the age model to include a hiatus at the previously identified position, but otherwise placed no constraints on its duration other than the duration must be strictly positive. Please see also our response to Dr. Blaauw's comment on hiatus prior distributions in our response to Referee Comment 1.

- Figure 6: Which of the two models in Figure 5 are we looking at here? Or is the result in Figure 6 identical for both models in Figure 5?

  This is the result of the *Meyers Model*. The EHA and periodograms for both models in figure 5 look more or less identical (note that the model medians are parallel), so we choose to only highlight one in figure 6.

- Line 466: Case 2 from the Cyclostratigraphic Intercomparison Project was designed by Christian Zeeden, not by Matthias Sinnesael. He should be acknowledged here.

  It was not our intent to exclude Dr. Zeeden. We requested the raw data from Dr. Sinnesael since he is the first author on the Cyclostratigraphic Intercomparison Project paper. We have updated the acknowledgements to say "We thank Dr. Matthias Sinnesael for providing, and Dr. Christian Zeeden for developing, the Cyclostratigraphy Intercomparison Project CIP2 data used for model testing."

Berger, A., Loutre, M.-F., and Laskar, J.: Stability of the astronomical frequencies over the Earth's history for paleoclimate studies, Science, 255, 560–566, 1992.

Bronk Ramsey, C.: Dealing With Outliers And Offsets In Radiocarbon Dating, Radiocarbon, 51, 1023–1045, https://doi.org/doi:10.1017/S0033822200034093, 2009.

Huybers, P. and Wunsch, C.: Obliquity pacing of the late Pleistocene glacial terminations, Nature, 434, 491–494, 2005.

Kuhnt, W., Nederbragt, A., and Leine, L.: Cyclicity of Cenomanian-Turonian organic-carbon-rich sediments in the Tarfaya Atlantic coastal basin (Morocco), Cretaceous Research, 18, 587–601, 1997.

Laskar, J., Robutel, P., Joutel, F., Gastineau, M., Correia, A. C., and Levrard, B.: A long-term numerical solution for the insolation quantities of the Earth, Astronomy & Astrophysics, 428, 261–285, 2004.

Laskar, J., Fienga, A., Gastineau, M., and Manche, H.: La2010: A new orbital solution for the long-term motion of the Earth, Astronomy & Astrophysics, 532, A89, 2011.

Lisiecki, L. E. and Raymo, M. E.: A Pliocene-Pleistocene stack of 57 globally distributed benthic $\delta$18O records, Paleoceanography, 20, 2005.

Malinverno, A., Erba, E., and Herbert, T.: Orbital tuning as an inverse problem: Chronology of the early Aptian oceanic anoxic event 1a (Selli Level) in the Cismon APTICORE, Paleoceanography, 25, 2010.

Meyers, S. R.: Cyclostratigraphy and the problem of astrochronologic testing, Earth-Science Reviews, 2019.

Meyers, S. R. and Sageman, B. B.: Detection, quantification, and significance of hiatuses in pelagic and hemipelagic strata, Earth and Planetary Science Letters, 224, 55–72, 2004.

Meyers, S. R. and Sageman, B. B.: Quantification of deep-time orbital forcing by average spectral misfit, American Journal of Science, 307, 773–792, 2007.

Meyers, S. R., Sageman, B. B., and Hinnov, L. A.: Integrated Quantitative Stratigraphy of the Cenomanian-Turonian Bridge Creek Limestone Member Using Evolutive Harmonic Analysis and Stratigraphic Modeling, Journal of Sedimentary Research, 71, 628–644, 2001.

Meyers, S. R., Siewert, S. E., Singer, B. S., Sageman, B. B., Condon, D. J., Obradovich, J. D., Jicha, B. R., and Sawyer, D. A.: Intercalibration of radioisotopic and astrochronologic time scales for the Cenomanian-Turonian boundary interval, Western Interior Basin, USA, Geology, 40, 7–10, 2012a.

Meyers, S. R., Sageman, B. B., and Arthur, M. A.: Obliquity forcing of organic matter accumulation during Oceanic Anoxic Event 2, Paleoceanography, 27, 2012b.

Michel, L. A., Schmitz, M. D., Tabor, N. J., Moontañez, I. P., and Davydov, V. I.: Reply to the comment on "Chronostratigraphy and paleoclimatology of the Lodève Basin, France: Evidence for a pan-tropical aridification event across the Carboniferous–Permian boundary" by Michel et al., (2015). Palaeogeography, Palaeoclimatology, Palaeoecology 430, 118–131, Palaeogeography Palaeoclimatology Palaeoecology, 441, 1000–1004, https://doi.org/doi:10.1016/j.palaeo.2015.10.023, 2016.

R Core Team: R: A Language and Environment for Statistical Computing, 2023.

Sageman, B. B., Rich, J., Arthur, M. A., Birchfield, G., and Dean, W.: Evidence for Milankovitch periodicities in Cenomanian-Turonian lithologic and geochemical cycles, Western Interior USA, Journal of Sedimentary Research, 67, 286–302, 1997.

Sageman, B. B., Rich, J., Arthur, M. A., Dean, W. E., Savrda, C. E., and Bralower, T. J.: Multiple Milankovitch Cycles in the Bridge Creek imestone (Cenomanian-Turonian), Western Interior Basin,

in: Stratigraphy and Paleoenvironments of the Cretaceous Western Interior Seaway, U.S.A, edited by: Dean, W. E. and Arthur, M. A., Special Publications of SEPM, 153–171, 1998.

Schmitz, M. D. and Kuiper, K. F.: High-Precision Geochronology, Elements, 9, 25–30, 2013.

Sinnesael, M., De Vleeschouwer, D., Zeeden, C., Batenburg, S. J., Da Silva, A.-C., de Winter, N. J., Dinarès-Turell, J., Drury, A. J., Gambacorta, G., and Hilgen, F. J.: The Cyclostratigraphy Inter-comparison Project (CIP): Consistency, merits and pitfalls, Earth-Science Reviews, 199, 102965, 2019.

Wallmann, K., Flögel, S., Scholz, F., Dale, A. W., Kemena, T. P., Steinig, S., and Kuhnt, W.: Periodic changes in the Cretaceous ocean and climate caused by marine redox see-saw, Nature Geoscience, 12, 456–461, 2019.

Waltham, D.: Milankovitch period uncertainties and their impact on cyclostratigraphy, Journal of Sedimentary Research, 85, 990–998, 2015.

Zeeden, C., Meyers, S. R., Lourens, L. J., and Hilgen, F. J.: Testing astronomically tuned age models, Paleoceanography, 30, 369–383, 2015.